# Multimodal Latent Language Modeling with Next-Token Diffusion

**Yutao Sun** [* 1]   **Hangbo Bao** [* 2]   **Wenhui Wang** [* 2]   **Zhiliang Peng** [* 2]   **Li Dong** [* 2]   **Shaohan Huang** [2]   **Yaoyao Chang** [2]
**Jianyong Wang** [1]   **Furu Wei** [2]

## Abstract

Multimodal generative models require a unified approach to handle both discrete data (e.g., text and code) and continuous data (e.g., image, audio, video). In this work, we propose Latent Language Modeling (LatentLM), which seamlessly integrates continuous and discrete data using causal Transformers. Specifically, we employ a variational autoencoder (VAE) to represent continuous data as latent vectors and introduce next-token diffusion for autoregressive generation of these vectors. Additionally, we develop $\sigma$-VAE to address the challenges of error accumulation and collapsed variance, which is crucial for autoregressive modeling. Extensive experiments demonstrate the effectiveness of LatentLM across various modalities. In image generation, LatentLM is competitive with or outperforms DiT-style baselines under matched unified settings. When integrated into multimodal large language models, LatentLM provides a general-purpose interface that unifies multimodal generation and understanding. Experimental results show that LatentLM achieves favorable performance compared to Transfusion and vector quantized models in the setting of scaling up training tokens. In text-to-speech synthesis, LatentLM outperforms the state-of-the-art VALL-E 2 model in speaker similarity and robustness, while requiring $10\times$ fewer decoding steps. The results establish LatentLM as a highly potential approach to advance large multimodal models.

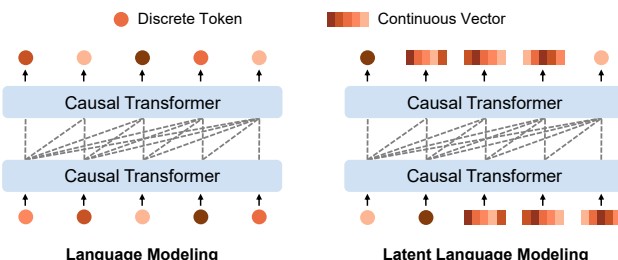

*Figure 1.* Latent Language Modeling (LatentLM) seamlessly handles continuous (e.g., image, audio, video) and discrete (e.g., text and code) data using causal Transformers. We introduce next-token diffusion to autoregressively generate the latent vectors one by one. The proposed method provides a general-purpose interface that unifies multimodal generation and understanding.

## 1. Introduction

Multimodal generative models require a unified method to process both discrete data (e.g., text, code) and continuous data (e.g., video, audio, robot actions). While pipeline-based systems exist, they lose information and cannot be optimized end-to-end. Research on natively handling both data types in multimodal large language models (MLLMs) has followed three main paths. First, continuous data can be quantized into discrete codes using VQ-VAE (van den Oord et al., 2017; Esser et al., 2021), making them compatible with autoregressive LLMs (Ramesh et al., 2021; Wang et al., 2023a; Team, 2024). Although simple, this approach creates a restrictive bottleneck, where lossy tokenization degrades both generation and understanding quality. Second, one can use diffusion models (Ho et al., 2020; Rombach et al., 2022; Bao et al., 2023b; Tang et al., 2023) for MLLMs. However, these models typically rely on bidirectional attention, making them fundamentally incompatible with the efficient, causal, autoregressive paradigm of LLMs that leverages KV cache. Frameworks like Transfusion (Zhou et al., 2024) attempt to bridge this gap, but still require separate objectives and cannot perform joint understanding and generation within a single forward pass, as the noisy inputs required for diffusion training interfere with the understanding tasks. The third path, masked autoregressive (Li et al., 2024) (MAR) models, offers strong generation by also leveraging bidirectional attention, but consequently shares the same incompatibility with causal LLMs and incurs high computational costs.

---

[*]Equal contribution [1]Tsinghua University [2]Microsoft Research. Correspondence to: Furu Wei <fuwei@microsoft.com>, Jianyong Wang <jianyong@tsinghua.edu.cn>.

In this work, we propose latent language modeling (LatentLM) to transcend the above trade-offs. LatentLM provides a single, causal, autoregressive architecture that supports both high-fidelity continuous generation and deep multimodal understanding without compromise. Specifically, we represent continuous data as high-fidelity latent vectors using a variational autoencoder (VAE). We then introduce next-token diffusion, where a lightweight diffusion head autoregressively predicts these latent vectors, conditioned on the causal Transformer's hidden states. For discrete data, the same Transformer backbone performs standard next-token prediction. Furthermore, to address the problem of error accumulation in continuous autoregressive generation, we propose $\sigma$-VAE, which maintains the variance of the latent space to mitigate exposure bias and ensure robust, long-sequence generation.

As shown in Figure 1, LatentLM unifies the generation of discrete and continuous tokens under a single language modeling paradigm. The proposed method simplifies implementation by reusing the existing distributed training infrastructure of large language models. Another advantage is that LatentLM unifies generation and understanding with a general-purpose interface, which perceives and produces any combination of multimodal data, e.g., text, image, audio, video, and robot action data. Compared to quantizing continuous data, LatentLM achieves a higher token reduction ratio while preserving reconstruction quality.

We conduct experiments on image generation, multimodal large language models, and text-to-speech synthesis to show the flexibility and effectiveness of LatentLM across modalities. First, image generation on ImageNet (Deng et al., 2009) shows that LatentLM achieves competitive performance with the models based on diffusion (e.g., DiT (Peebles & Xie, 2023)) or discrete tokens. Second, we train multimodal large language models with text, image-text pairs, and interleaved data. LatentLM outperforms Transfusion (Zhou et al., 2024) and the model with vector-quantized image tokenizers, in terms of language modeling, text-to-image generation, and vision-language understanding metrics. We also scale up the number of training tokens and find that LatentLM has favorable scaling properties. Third, experimental results on text-to-speech synthesis show that LatentLM achieves better performance than previous systems. Because our tokenizer uses continuous representations, the token reduction ratio is much larger than previous vector-quantized tokenizers, which improves both the training and inference efficiency.

## 2. Latent Language Modeling

Latent language modeling (LatentLM) autoregressively perceives and generates multimodal sequences (with discrete and continuous data) in a unified way. As shown in Figure 2, the model is a causal Transformer, where the $t$-th token is predicted by conditioning on previous $t - 1$ tokens. Continuous data are generated by next-token diffusion (Section 2.1), where the diffusion head is used to produce continuous vectors for each position. In addition, discrete tokens are generated by next-token prediction, similar to conventional language modeling.

Specifically, let $x = x_1 \cdots x_N$ denote an input sequence of discrete and continuous tokens. For a discrete token, we use a lookup table to get its vector representation. For continuous data, variational autoencoder (VAE; Kingma & Welling 2014) is used as tokenizer to compress input data to latent vectors (Section 3). After obtaining the vector representations, we pack the input vectors into $X^0 = [\boldsymbol{x}_1, \cdots, \boldsymbol{x}_N] \in \mathbb{R}^{N \times d}$, where $d$ represents the hidden dimension of the model. $X^0$ is fed into a language model based on causal Transformer.

The model is stacked with $L$ Transformer (Vaswani et al., 2017) layers with LLaMA augmentation (Touvron et al., 2023). Causal masking is used for autoregressive generation. The input $X^0$ is further contextualized to obtain the output $X^L$, i.e., $X^l = \text{Decoder}(X^{l-1})$, $l \in [1, L]$. The output states of Transformer $[\boldsymbol{h}_1, \cdots, \boldsymbol{h}_N] = \text{RMSNorm}(X^L)$ are used to decode the predictions:

$$\text{Decode}(x_i|x_{<i}) = \begin{cases} \text{Sample}\left(P_d(x_i|x_{<i})\right) & x_i \in \mathcal{D} \\ \text{Diffusion}(\boldsymbol{h}_i) & x_i \in \mathcal{C} \end{cases}$$
$$P_d(x_i|x_{<i}) = \text{softmax}(\boldsymbol{h}_i W_v) \tag{1}$$

where $W_v \in \mathbb{R}^{d \times |\mathcal{V}|}$ is the $\text{softmax}$ classifier weight, $|\mathcal{V}|$ is the vocabulary size, $\mathcal{D}$ is discrete token set, $\mathcal{C}$ is continuous token set, and $\text{Sample}(\cdot)$ is a sampling algorithm (e.g., greedy decoding, and top-$p$ sampling). The $\text{Diffusion}(\cdot)$ head is described in Section 2.1, which decodes continuous vectors by conditioning on the hidden state $\boldsymbol{h}_i$. The latent vectors are generated autoregressively one by one, i.e., next-token diffusion. Then the VAE decoder is used to generate raw data from the predicted latent vectors.

### 2.1. Next-Token Diffusion

LatentLM autoregressively generates the continuous tokens. We use diffusion as the language model head for each continuous token. The diffusion head progressively refines and generates the latent vector $\boldsymbol{x}_i$ by conditioning on the hidden state $\boldsymbol{h}_i$. Then the predicted $\boldsymbol{x}_i$ is used as input for the next step of Transformer.

In our experiments, we use either denoising diffusion probabilistic model (DDPM) (Ho et al., 2020) or flow matching (Lipman et al., 2022) as our design choice. We use DDPM as an example to describe the details. Diffusion is formulated as two processes, i.e., the forward process gradually adds noise to the input, and the reverse process

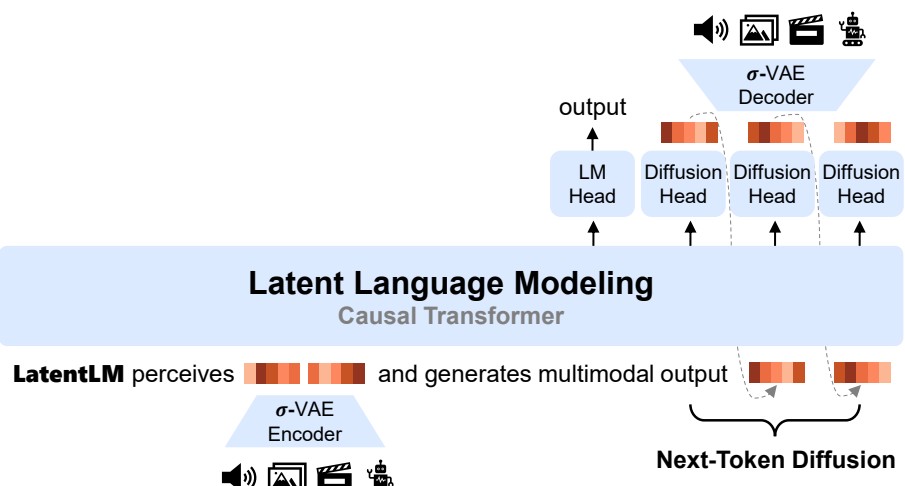

*Figure 2.* LatentLM unifies the modeling of continuous and discrete data with causal Transformers. We introduce $\sigma$-VAE (Section 3) to represent continuous data as latent vectors. We perform next-token diffusion (Section 2.1) to autoregressively predict the latent vectors one by one. The diffusion head generates vectors by conditioning on the output states of Transformer. The predicted vectors can be decoded to produce the final outputs.

learns to denoise step by step.

**Forward Process** Noise is introduced incrementally into the original vector in $T$ steps. Let $\boldsymbol{x}_i^0 = \boldsymbol{x}_i$ denote the original data and $\boldsymbol{x}_i^t$ the noisy version, where $t = 1, \cdots, T$. The Markov noise-addition process is defined as $q(\boldsymbol{x}_i^t|\boldsymbol{x}_i^{t-1}) = \mathcal{N}(\boldsymbol{x}_i^t; \sqrt{1-\beta_t}\boldsymbol{x}_i^{t-1}, \beta_t\boldsymbol{I})$, where Gaussian noise is injected in each step, $\beta_t$ follows a predefined noise schedule, and $\boldsymbol{I}$ is the identity covariance matrix. A nice property is that we can directly sample $\boldsymbol{x}_i^t$ from the original data $\boldsymbol{x}_i$ through:

$$\boldsymbol{x}_i^t = \sqrt{\overline{\alpha}_t}\boldsymbol{x}_i + \sqrt{1-\overline{\alpha}_t}\boldsymbol{\epsilon} \qquad (2)$$

where $\overline{\alpha}_t = \prod_{i=1}^{t}(1-\beta_i)$, and $\boldsymbol{\epsilon} \sim \mathcal{N}(0, \boldsymbol{I})$.

**Reverse Process** The diffusion model learns to reverse the noising process by training a network $p_\theta(\boldsymbol{x}_i^{t-1}|\boldsymbol{x}_i^t, \boldsymbol{h}_i)$ to predict the noise added at each step. DDPM learns a model $\boldsymbol{\epsilon}_\theta(\boldsymbol{x}_i^t, t, \boldsymbol{h}_i)$ to estimate the noise $\boldsymbol{\epsilon}$ (as described in the forward pass) of $\boldsymbol{x}_i^t$ in the $t$-th step, conditioning on the Transformer state $\boldsymbol{h}_i$. The model parameters are learned by minimizing the following loss:

$$\mathcal{L}_{\text{Diff}}(\boldsymbol{x}_i, \boldsymbol{h}_i) = \mathbb{E}_{\boldsymbol{x}_i, t, \boldsymbol{\epsilon}} \parallel \boldsymbol{\epsilon} - \boldsymbol{\epsilon}_\theta(\boldsymbol{x}_i^t, t, \boldsymbol{h}_i) \parallel^2 \qquad (3)$$

where $\boldsymbol{\epsilon}$ is the actual Gaussian noise.

**Head Architecture** We use a lightweight neural network as $\boldsymbol{\epsilon}_\theta(\cdot)$ in Equation (3), which is a residual architecture incorporating pre-RMSNorm (Zhang & Sennrich, 2019) and feedforward networks (Li et al., 2024). The network input is a vector that contains noise. The output is the predicted noise $\boldsymbol{\epsilon}_\theta(\cdot)$. We also utilize AdaLN-Zero (Peebles & Xie, 2023) which conditions on both the timestep $t$ and the Transformer output $\boldsymbol{h}_i$. This head processes a noised continuous vector and predicts the corresponding noise.

**Inference** The Transformer state $\boldsymbol{h}_i$ is used as the condition for diffusion head. The diffusion process iteratively denoises data. At first, a vector of pure Gaussian noise $\boldsymbol{x}_T$ is given. In each step, the predicted noise $\boldsymbol{\epsilon}_\theta(\boldsymbol{x}_i^t, t, \boldsymbol{h}_i)$ is used to produce $\boldsymbol{x}_{t-1}$ from $\boldsymbol{x}_t$, which also considers the noise schedule for scaling (Ho et al., 2020). In our experiments, we utilize DPM-Solver (Lu et al., 2022a;b) to accelerate the denoising process, significantly reducing the number of inference steps compared to the training phase.

### 2.2. Model Training and Inference

**Training** During training, we compute the token-level loss for training sequences. For discrete data, we use the standard language modeling objective to maximize the likelihood of data. Specifically, the loss is computed as $\mathcal{L}_{\text{LM}} = -\sum_{x,i} \log P_d(x_i|x_{<i})$, where the prediction probability is presented in Equation (1). For continuous data, the loss function $\mathcal{L}_{\text{Diff}}$ described in Equation (3) is used. The training objective is to minimize $\mathcal{L}_{\text{LM}} + \alpha\mathcal{L}_{\text{Diff}}$, where $\alpha$ is a hyperparameter. In practice, we sample multiple diffusion timesteps, typically four, for a single forward pass (Li et al., 2024). As the diffusion head is usually lightweight, reusing the computation of the Transformer backbone improves training efficiency while introducing minimal overhead.

**Inference** The decoding process is similar to that of standard causal Transformers, i.e., predicting the next token based on the generation history that has come before it. The tokens are produced following Equation (1). Notice that the Transformer backbone is computed in a single pass, and only the lightweight diffusion head requires multiple denoising steps. In addition, we use special tokens to indicate the switch between the language modeling head and the diffusion head. For instance, we use <BOD> to denote

the beginning of the diffusion head usage, and `<EOD>` to indicate the switch back to the language modeling head.

## 3. $\sigma$-VAE: The Keystone for Next-Token Diffusion

### 3.1. Background: Variational Autoencoder (VAE)

The tokenizer compresses continuous data into latent vectors. It is based on variational autoencoder (VAE; Kingma & Welling 2014), which encodes the input data into a latent space and then decodes it back to the original space. Let $x$ denote the continuous input and $z$ the compressed vector representations. VAEs maximize the evidence lower bound of log-likelihood $\log p(x)$ via:

$$\max \ \mathbb{E}_{q_\phi(z|x)}\left[\log p_\psi(x|z)\right] - \mathcal{D}_{\mathrm{KL}}\left[q_\phi(z|x) \parallel p(z)\right] \quad (4)$$

where the encoder $q_\phi(z|x)$ encodes input $x$ to latent vectors $z$, the decoder $p_\psi(x|z)$ reconstructs data by conditioning on $z$, and the KL term encourages that the latent space follows a Gaussian prior.

### 3.2. The Ineffectiveness of Conventional VAEs for Autoregressive Modeling

Figure 5 shows that conventional VAEs perform poorly for autoregressive modeling. We explain why conventional VAEs are not suitable for this purpose as follows.

**Training-Inference Mismatch** The issue relates to exposure bias in autoregressive models, which becomes more severe in continuous latent spaces. Consider a latent token $x$ whose VAE posterior is modeled as $\mathcal{N}(x, \sigma^2)$. During training with teacher forcing, the model always observes samples from $\mathcal{N}(x, \sigma^2)$. However, during inference, it uses its own generated outputs as inputs. These generated latent vectors come from a different distribution, call it $\mathcal{N}(x, \sigma_{\mathrm{gen}}^2)$, and in practice $\sigma_{\mathrm{gen}}^2$ can be larger, due to compounding generation errors. This mismatch between the training and inference distributions can lead to degradation over time, especially in long-form autoregressive generation. In traditional VAEs, where the variance may vary across tokens (and in some cases be extremely small), this divergence can push the model into out-of-distribution regions during inference.

**Collapsed Variance** The VAE models used in previous diffusion models typically perform more like an autoencoder, i.e., with collapsed $\sigma$, as shown in the statistics of Figure 5. This leads to inference degradation when diffusion is combined with autoregressive decoding, as explained above. Although the variance is not directly fed to the model, the variance affects the teacher-forcing inputs fed into the autoregressive training. Larger $\sigma$ of VAE makes the model more robust to exposure bias.

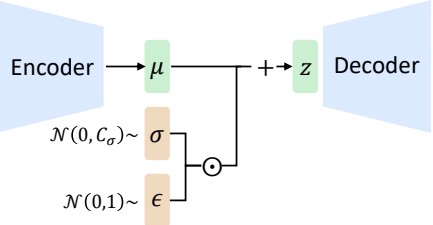

*Figure 3.* $\sigma$-VAE uses a fixed variance for the latent space. It avoids variance collapse and makes LatentLM more robust to exposure bias during autoregressive generation. $\sigma$ is a scalar that is sampled from $\mathcal{N}(0, C_\sigma)$ for each example.

### 3.3. Solution: $\sigma$-VAE

As shown in Figure 3, we propose $\sigma$-VAE to prevent variance collapse by enforcing a fixed variance in the latent space. The reconstruction pass is computed as:

$$\begin{aligned}
\mu &= \mathrm{Encoder}_\phi(x) \\
z &= \mu + |\sigma| \odot \epsilon, \text{ where } \epsilon \sim \mathcal{N}(0, 1), \ \sigma \sim \mathcal{N}(0, C_\sigma) \\
\hat{x} &= \mathrm{Decoder}_\psi(z)
\end{aligned}$$

where $\sigma$ is a scalar, $C_\sigma$ is a hyperparameter, $\mathrm{Encoder}_\phi(\cdot)$ and $\mathrm{Decoder}_\psi(\cdot)$ are learnable models. The input $x$ is fed into the encoder to obtain $\mu$. The re-parameterization trick is used for $z$. The variance $\sigma$ is fixed across channels, and is sampled from $\mathcal{N}(0, C_\sigma)$ for each example. Then $z$ is fed into the decoder for reconstruction.

The training objective of $\sigma$-VAE is:

$$\mathrm{minimize} \quad \|\hat{x} - x\|_2^2 + \beta \|\mu\|_2^2$$

where the first term is the reconstruction error, and $\beta$ controls the trade-off between reconstruction quality and adherence to the prior distribution (Higgins et al., 2016).

**Advantages** The proposed $\sigma$-VAE addresses the above issues (described in Section 3.2) by explicitly lower-bounding the latent variance. This makes the model robust to generation error by ensuring the inference-time distribution stays within the distributional support seen during training. In practice, we observe that after training, the condition $\sigma_{\mathrm{gen}} < \sigma_{\mathrm{train}}$ tends to hold in most tokens, effectively converting an out-of-distribution generalization problem into an in-distribution one, thereby significantly improving stability in autoregressive inference.

## 4. Experiments

We evaluate LatentLM through multiple dimensions to thoroughly assess its effectiveness and scalability potential. We conduct experiments on various tasks, i.e., image generation, multimodal large language models, and text-to-speech synthesis.

| Type | Model | #Params | #Epochs | FID↓ | IS↑ |
|------|-------|---------|---------|------|-----|
| *Non-Causal-Masking Generation* | | | | | |
| Diffusion | LDM-4 (Rombach et al., 2022) | 400M | — | 3.60 | 247.7 |
| | DiT-XL/2 (Peebles & Xie, 2023) | 675M | 400 | 2.27 | 278.2 |
| | U-ViT-H/2 (Bao et al., 2023a) | 501M | 400 | 2.29 | 263.9 |
| Masked Generative | MaskGIT (Chang et al., 2022) | 227M | 300 | 4.02 | 355.6 |
| | MAR-L (Li et al., 2024) | 479M | 800 | 1.78 | 296.0 |
| *Causal-Masking Generation* | | | | | |
| Causal-Discrete | VQGAN (Esser et al., 2021) | 1.4B | 240 | 5.20 | 280.3 |
| | ViT-VQGAN (Yu et al., 2021) | 1.7B | 240 | 3.04 | 227.4 |
| | LlamaGen-XL (Sun et al., 2024a) | 775M | 300 | 2.62 | 244.1 |
| | LlamaGen-XXL (Sun et al., 2024a) | 1.4B | 300 | 2.34 | 253.9 |
| Causal-Continuous | GIVT-Causal-L+A (Tschannen et al., 2023) | 1.67B | 500 | 2.59 | — |
| | LatentLM-L (This Work) | 479M | 400 | 2.24 | 253.8 |

*Table 1.* Image generation results on ImageNet (Deng et al., 2009). We evaluate FID (Heusel et al., 2017) and IS (Salimans et al., 2016). LatentLM achieves competitive performance, especially compared with other causal-masking image generation models.

## 4.1. Image Generation: Scalable Autoregressive Modeling

The image generation experiments are conducted on ImageNet (Deng et al., 2009). Given a category, the goal is to generate the corresponding images. First, we systematically benchmark our model against state-of-the-art baselines to demonstrate the advantages of next-token diffusion. We also investigate the scalability of our approach by evaluating it with larger model sizes and higher resolutions. Furthermore, we compare the design choices of $\sigma$-VAE tokenizers. Finally, we assess the inference efficiency to highlight the practical deployment benefits of our method.

Our image generation experiments use raster-order causal factorization. The tokenizer-generator alignment issue studied here is orthogonal to the specific order, and alternative factorization strategies may further improve generation quality. In our main large-scale experiments, we use $\sigma$-VAE with Gaussian variance sampling rather than a fixed variance. Unless otherwise specified, we set $C_\sigma = 0.75$.

### 4.1.1. SYSTEM EVALUATION

**Setup** We scale up model size and number of training steps. We set the Transformer's hidden size to 1024 and the number of layers to 32. The intermediate dimension of feedforward networks is 2730. The diffusion head has six layers. We use the AdamW (Loshchilov & Hutter, 2019) optimizer with $\beta = (0.9, 0.98)$. We use a cosine learning rate schedule with the maximal value of 5e-4 and 100 warmup steps. The weight decay is set to 0.1. We train models with 250,000 steps with batch size of 2048. The number of training epochs is about 400. Classifier-free guidance (Ho & Salimans, 2022) is set to 1.65. As shown in Table 1, the model configurations have been aligned with those of previous models to ensure fair comparisons. The training details of $\sigma$-VAE are presented in Appendix I.2.

Table 1 presents a comprehensive comparison between LatentLM and various image generation methods. These methods can be categorized into two main groups: (1) non-causal-masking models, including image-level diffusion models (LDM (Rombach et al., 2022), DiT (Peebles & Xie, 2023), U-ViT (Bao et al., 2023a)) and masked generative models (MaskGIT (Chang et al., 2022), MAR (Li et al., 2024)); and (2) causal-masking models, comprising discrete-token generation approaches (VQGAN (Esser et al., 2021), ViT-VQGAN (Yu et al., 2021), LlamaGen (Sun et al., 2024a)) and continuous autoregressive generation methods (GIVT-Causal; Tschannen et al. 2023).

**Results** As shown in Table 1, LatentLM achieves competitive performance compared to previous work. Notice that non-causal-masking models typically require iterative forward computation during inference. Consequently, the inference FLOPs of non-causal-masking models tend to be larger due to multiple forward passes. Moreover, models using continuous representations typically outperform those using discrete code, even though LatentLM-L has fewer parameters. Among the methods, MAR (Li et al., 2024) and GIVT (Tschannen et al., 2023) are the most relevant. In comparison, MAR uses a bidirectional Transformer to implement masked autoregressive modeling, instead of causal Transformer, which renders MAR unable to reuse key-value caches for multiple forward passes. Furthermore, unifying MAR and language modeling in multimodal models remains challenging due to their distinct modeling approaches. In contrast, Section 4.2 shows that our approach can be naturally applied to multimodal large language models. In addition, GIVT directly predicts latent vectors of VAEs with Gaussian mixture models. The main difference is that LatentLM integrates diffusion into causal Transformers, which tends to offer more powerful modeling expressivity. The results also indicate that our approach outperforms GIVT with a smaller model size and fewer training epochs.

### 4.1.2. Scalability

We compare the scalability properties of Diffusion Transformer (DiT; Peebles & Xie 2023) and LatentLM, in terms of model size, and image resolution.

**Setup** In order to be consistent with LatentLM, we also augment DiT with RMSNorm (Zhang & Sennrich, 2019) and SwiGLU (Ramachandran et al., 2017; Shazeer, 2020). All models were trained with 75k steps, i.e., approximately 120 epochs, for scaling experiments. Classifier-free guidance (Ho & Salimans, 2022) is set to 1.75 during inference. Detailed hyperparameters are presented in Appendix I.1.

**Scaling Image Resolution** As shown in Table 2, we conduct experiments at a resolution of 384, training a 1.82B model for 100k steps. The results of 384-pixel resolution show significant improvements over the 256 when using classifier-free guidance (Ho & Salimans, 2022). The improvement stems from the richer details and additional information captured in the tokenizer with higher resolutions. Moreover, increasing resolution leads to longer sequences, which scales the decoding computation up.

| Resolution | FID↓ |
|---|---|
| $256 \times 256$ | 3.19 |
| $384 \times 384$ | **2.51** |

*Table 2.* FID-50k (Heusel et al., 2017) results of scaling up image resolution.

**Scaling Model Size** As shown in Figure 4, we trained models of varying sizes, i.e., 455M, 1.03B, 1.82B, 3.68B. LatentLM consistently outperforms DiT models. The results demonstrate our approach's effective scaling properties in terms of model size.

### 4.1.3. Effects of Tokenizer

As shown in Figure 5, we analyze the effects of $\sigma$-VAE tokenizers with various configurations. We evaluate their performance in both the DiT and LatentLM frameworks. Specifically, we train the $\sigma$-VAE tokenizers with different variance. To simplify the analysis, we use fixed variance values $\sigma$, rather than sampling them from $\mathcal{N}(0, C_\sigma)$. We make ablation on variance sampling in Appendix B. More comparison details are presented in Appendix I.3.

Figure 5 presents the FID-50K scores of DiT and LatentLM using tokenizers with different variance. The "stars" in the figure represent tokenizers that were tuned for previous latent diffusion models (Rombach et al., 2022), which usually have a small variance, i.e., being more like an autoencoder instead of VAE. The other "dots" are $\sigma$-VAE with fixed variance. We summarize the findings as follows:

**The tokenizers tuned for previous image-level diffusion models are ineffective for LatentLM.** For LatentLM, the "stars" (in Figure 5) perform significantly worse than the others that have larger tokenizer variances. The results indicate that directly adopting tokenizer configurations from previous diffusion models is suboptimal for LatentLM. The tokenizers with small variances are not robust to autoregressive error (Tschannen et al., 2023).

**LatentLM favors tokenizers with larger variances.** For the example without classifier-free guidance (i.e., CFG=1.0 in Figure 5), LatentLM improves monotonically with increased variance. In contrast, the choice of variance is not critical for DiT models. The analysis highlights the advantage of $\sigma$-VAE, whose variance is easily controllable. So we recommend to use re-trained $\sigma$-VAE as tokenizers for LatentLM, rather than directly using previous ones.

We further observed that training with sampled latents as inputs while predicting the posterior mean latent gives faster early optimization, but becomes worse after full convergence. This suggests that reducing target stochasticity alone does not remove the train-inference mismatch, because recursively generated latents still move away from the training input distribution over rollout depth.

## 4.2. Multimodal LLMs: Unified Understanding and Generation

We train multimodal large language models with LatentLM for unified understanding and generation. In this section, we focus on vision-language models. By unifying next-token prediction and diffusion, the model can seamlessly handle interleaved image-text data, text-only data, and image-text pairs. The proposed method simplifies the multimodal training and inference processes, providing a natural interface for future in-context learning, instruction following, and multimodal dialogue. Moreover, unified modeling enables new capabilities. For example, we can edit or generate images by conditioning on text and multiple images.

We use three types of data in the training stage: text-only data, image-text pair data, and interleaved text-image data. The mix-up ratio is 2:1:1. We train a 1.3B-size Transformer as the backbone. The training sequence length is 4096. The batch size is 4M tokens. We train the model with 50,000 steps (i.e., 200B tokens) for comparison. More training details are described in Appendix I.4.

We compare LatentLM with Transfusion (Zhou et al., 2024), and vector quantized models (VQ-MLLM; i.e., the models using vector quantized image tokenizers). Specifically, Transfusion shares Transformer weights for autoregressive language modeling and image-level diffusion, which uses bidirectional iterative denoising for images and causal masking for text. Moreover, VQ-MLLM uses VQ-VAE (van den Oord et al., 2017; Esser et al., 2021) as the tokenizer for images, where images are compressed to discrete code. We use the VQ-VAE tokenizer open-sourced by LlamaGen (Sun

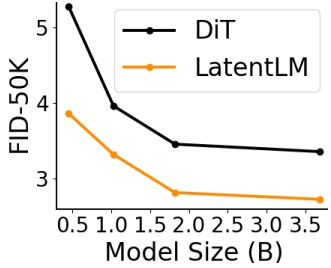
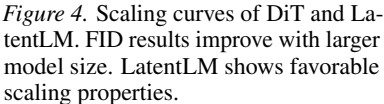

Figure 4. Scaling curves of DiT and La-tentLM. FID results improve with larger model size. LatentLM shows favorable scaling properties.

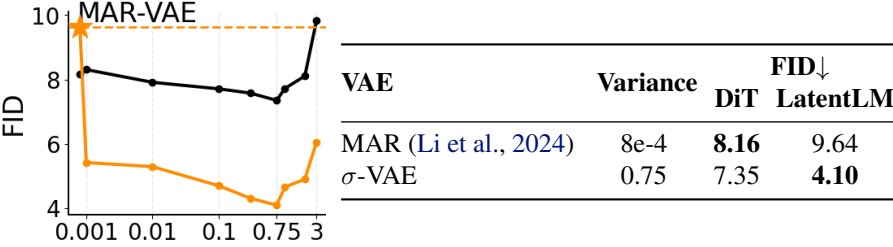

| VAE | Variance | FID↓ | |
| --- | --- | --- | --- |
| | | DiT | LatentLM |
| MAR (Li et al., 2024) | 8e-4 | **8.16** | 9.64 |
| $\sigma$-VAE | 0.75 | 7.35 | **4.10** |

Figure 5. FID (Heusel et al., 2017) scores of Diffusion Transformer (DiT) and LatentLM on ImageNet. The "star" (MAR-VAE) represents the tokenizer tuned for previous diffusion models (Li et al., 2024), which are ineffective for LatentLM. The results indicate that LatentLM favors tokenizers with larger variances, highlighting the importance of $\sigma$-VAE.

| Model | Text | Text-to-Image | | Image-to-Text | |
| --- | --- | --- | --- | --- | --- |
| | Valid PPL↓ | FID↓ | CLIP↑ | MS-COCO↑ | VQAv2↑ |
| VQ-MLLM | 2.79 | 16.92 | **29.33** | 37.4 | 30.19 |
| Transfusion (Zhou et al., 2024) | 2.74 | 16.10 | 28.66 | 43.4 | 35.36 |
| LatentLM | **2.73** | **14.54** | 28.75 | **54.5** | **38.72** |

Table 3. Results of multimodal large language models on text language modeling, image-to-text, and text-to-image generation. We compare with Transfusion (Zhou et al., 2024) and vector quantized models (VQ-MLLM; i.e., using discrete code to represent images). "PPL" is perplexity. CLIP (Radford et al., 2021) score measures the similarity. We report CIDEr (Vedantam et al., 2015) score for MS-COCO (Lin et al., 2014b) and accuracy for VQAv2 (Goyal et al., 2017).

et al., 2024a) in VQ-MLLM. We use the same training configuration and tokenizer settings for comparison. To align the number of parameters, we use a 6-layer ViT as the image head of Transfusion, wchih contains 327.2M parameters, larger than a 6-layer MLP-only stack with 226.5M parameters and also larger than the image component used in our method.

We vary the diffusion loss weight from 1 to 10 and observe that the cross-entropy loss remains nearly unchanged. The diffusion loss also saturates when the weight increases from around 5 to 10. We therefore follow the Transfusion-style choice and set the weight around 5, where the numerical scales of the two losses are comparable.

**Language Modeling** Table 3 presents the evaluation results on language modeling, text-to-image generation, and multimodal understanding. First, LatentLM achieves a better perplexity in language modeling. The results indicate that our method tends to better share knowledge between modalities with less conflicts. The similarity between next-token prediction and next-token diffusion also benefits the unified modeling. We further evaluate language-only tasks in Appendix E showing the advantage of LatentLM.

**Text-to-Image Generation** Then we evaluate text-to-image generation on MS-COCO (Lin et al., 2014a). Table 3 shows that LatentLM achieves lower FID scores, i.e., better generation quality. The trend is also consistent with Table 1, where Transfusion is aligned with DiT, and VQ-

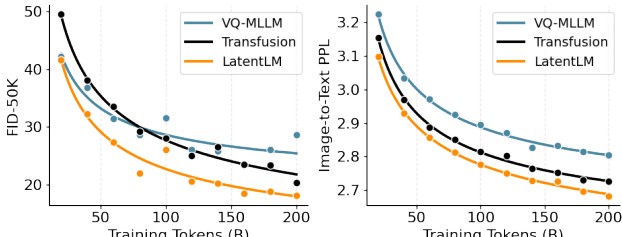

(a) Text-to-image FID (Heusel et al., 2017).

(b) Image-to-text validation perplexity.

Figure 6. We scale up the number of training tokens for multimodal large language models. LatentLM outperforms vector quantized models (VQ-MLLM) and Transfusion (Zhou et al., 2024) for both text-to-image and image-to-text generation. The FID scores are evaluated on MS-COCO (Lin et al., 2014b).

MLLM with LlamaGen. In addition, Figure 6a presents the scaling curves in terms of the number of training tokens, where LatentLM consistently achieves better FID scores. It is worth noting that the performance of VQ-MLLM seems saturated compared to the other methods. Figure 12 also shows several text-to-image samples of LatentLM.

**Image-to-Text Generation** Table 3 reports image captioning on MS-COCO (Lin et al., 2014a) and visual question answering on VQAv2 (Goyal et al., 2017). LatentLM achieves better performance in both multimodal understanding tasks. Compared to VQ-MLLM, the continuous representations used by Transfusion and LatentLM are more lossless than discrete code. Compared to Transfusion, LatentLM keeps training and inference consistent, rather than adding noise

| System | Frame Rate Length/s ↓ | Ref Utterance as Prompt | | | 3s Prefix as Prompt | | |
|---|---|---|---|---|---|---|---|
| | | SIM↑ | WER-C↓ | WER-H↓ | SIM↑ | WER-C↓ | WER-H↓ |
| Ground Truth | - | 0.779 | 1.6 | 2.2 | 0.668 | 1.6 | 2.2 |
| VALL-E 2 (Chen et al., 2024) | 75 | 0.643 | 1.5 | 2.4 | 0.504 | 1.6 | 2.3 |
| Voicebox (Le et al., 2023) | 100 | 0.662 | - | 1.9 | **0.593** | - | **2.0** |
| MELLE (Meng et al., 2024) | 62 | 0.625 | 1.5 | 2.1 | 0.508 | 1.5 | **2.0** |
| LatentLM | 15 | **0.697** | **1.2** | 1.8 | 0.571 | **1.4** | **2.0** |
| LatentLM | 7.5 | 0.656 | **1.2** | **1.7** | 0.532 | 1.6 | 2.3 |
| LatentLM | 3.75 | 0.598 | 1.7 | 2.3 | 0.467 | 3.1 | 4.5 |

*Table 4.* LatentLM outperforms previous systems on zero-shot speech synthesis in both settings. Frame rate indicates how many autoregressive steps to generate one second of speech. Moreover, the number of decoding steps is much less than others, achieving faster inference speed. The results are reported on LibriSpeech test-clean set. The WER-H and SIM results of VALL-E 2 using 3s prefix as prompt are from (Meng et al., 2024). The evaluation metrics are described in Appendix I.5.3.

| Tokenizer | $N_q$↓ | Frame Rate ↓ | Comp. Ratio↑ | Mel Dist.↓ | PESQ↑ | STOI↑ | VISQOL↑ | UTMOS↑ |
|---|---|---|---|---|---|---|---|---|
| DAC (Shechtman & Dekel, 2024) | 2 | 75 | 160 | 0.916 | 2.269 | 0.896 | 3.981 | 3.297 |
| WavTokenizer (Ji et al., 2024) | 1 | 75 | 320 | 0.871 | 2.266 | 0.891 | 4.120 | 3.432 |
| Mimi (Défossez et al., 2024) | 8 | 12.5 | 240 | 0.987 | 3.217 | 0.946 | 4.332 | 3.375 |
| Mimi (Défossez et al., 2024) | 4 | 12.5 | 480 | 1.458 | 1.568 | 0.826 | 3.390 | 2.652 |
| WavTokenizer (Ji et al., 2024) | 1 | 40 | 600 | 1.037 | 1.670 | 0.834 | 3.782 | 3.053 |
| $\sigma$-VAE$_{32}$ | 1 | 15 | 1600 | 0.813 | 2.724 | 0.926 | 4.268 | 3.491 |
| $\sigma$-VAE$_{64}$ | 1 | 7.5 | 3200 | **0.798** | **2.756** | **0.929** | **4.289** | **3.505** |
| $\sigma$-VAE$_{128}$ | 1 | 3.75 | 6400 | 0.852 | 2.533 | 0.916 | 4.165 | 3.460 |

*Table 5.* The $\sigma$-VAE tokenizers obtain competitive reconstruction quality while having high token reduction ratio. We report results on the LibriTTS test-other set and compare with the tokenizers whose token reduction ratio is larger than 100. "$N_q$" represents the number of quantizers. We define the token reduction ratio as the audio sample rate divided by $N_q$ and the frame rate. "$\sigma$-VAE$_{32}$" denotes that the latent dimension of the tokenizer is 32. The evaluation metrics are described in Appendix I.5.3.

to input images during training. Figure 6b presents text perplexity on the image-to-text validation data. The results are also consistent with those in Table 3.

### 4.3. Text-to-Speech Synthesis: Higher token reduction ratio, Fewer Decoding Steps

We apply LatentLM to text-to-speech synthesis (TTS). Due to continuous representations, $\sigma$-VAE achieves superior reconstruction results with a significantly higher token reduction ratio and lower frame rate than previous speech tokenizers (Défossez et al., 2022; Kumar et al., 2023; Shechtman & Dekel, 2024; Ji et al., 2024; Défossez et al., 2024). LatentLM outperforms the state-of-the-art VALL-E 2 (Chen et al., 2024) model on both speaker similarity score and robustness while requiring 10× fewer decoding steps.

We follow the settings of VALL-E 2 (Chen et al., 2024). The training data are Libriheavy (Kang et al., 2024), which includes 50k hours of speech from approximately 7k different speakers. The Transformer backbone has about 300M parameters. The diffusion head contains three layers of feed-forward networks. $\sigma$-VAE employ a convolutional architecture that supports streaming encoding and decoding. We train variants with token reduction ratios of 1600, 3200, and 6400. More training details are presented in Appendix I.5.

**System Evaluation** Table 4 shows zero-shot TTS results on the LibriSpeech test-clean set. We evaluate the synthesis quality under two distinct settings: (1) using a reference utterance from the same speaker as the prompt, and (2) evaluating speech continuation by using the first 3 seconds of speech as the prompt.

Our model, operating at a frame rate of 15 (i.e., generating 1 second of speech in 15 autoregressive steps), surpasses previous state-of-the-art methods when using a same-speaker reference utterance as the prompt. LatentLM with a frame rate of 7.5 achieves superior performance compared to the neural codec language model VALL-E 2 (Chen et al., 2024), while requiring an order of magnitude (10×) fewer autoregressive inference steps. Moreover, LatentLM eliminates the need for the non-autoregressive (NAR) model employed in VALL-E 2, resulting in improved computational efficiency. Even at a lower frame rate of 3.75, LatentLM maintains competitive performance. The higher token reduction ratio reduces the sequence length, which in turn greatly accelerates the decoding speed.

**Evaluating the Quality of Tokenizers** Table 5 compares $\sigma$-VAE and other codec models on the LibriTTS test-other set. $\sigma$-VAE achieves better reconstruction quality in a token reduction ratio of 1600× compared to Encodec (40×;

Défossez et al. 2022), DAC (160×; Shechtman & Dekel 2024), WavTokenizer (320×; Ji et al. 2024), and Mimi (480×; Défossez et al. 2024). Notably, as we further increase the token reduction ratio, the reconstruction quality does not deteriorate significantly. At a token reduction ratio of 6400, the resulting sequence length when used in a language model is already comparable to BPE tokenization (Sennrich et al., 2015), approaching a 1:1 ratio.

## 5. Related Work

**Continuous Autoregressive Generation**   Using discrete tokens for autoregressive generation has been a common approach (Esser et al., 2021; Yu et al., 2021; Sun et al., 2024a). In the context of continuous token generation, recent work has explored alternative mechanisms. For example, GIVT (Tschannen et al., 2023) introduces the use of a VAE head or adapter to predict continuous tokens, while MAR (Li et al., 2024) proposes the use of a diffusion head for masked generation. MAR's primary contribution is its masked bidirectional attention model; its causal attention baseline was shown to underperform, a finding which our work revisits. LatentLM combines the strengths of diffusion as a powerful predictive model with error-tolerant $\sigma$-VAE, demonstrating that a purely causal autoregressive approach can be highly effective and MLLM-compatible.

**Unified Multimodal LLM**   Previous unified multimodal LLM approaches can be broadly classified into two types. The first, based on discrete token models (Team, 2024; Liu et al., 2024; Wu et al., 2024; Wang et al., 2024), usually sacrifices some multimodal capability with vector quantization. These models excel at text tasks, but struggle with complex multimodal interactions. The second approach integrates LLM backbones for image-level diffusion (Zhou et al., 2024; Xie et al., 2024; Deng et al., 2025), but this is incompatible with autoregressive generation and increases inference complexity. Moreover, noisy image input during pretraining reduces training efficiency. In contrast, LatentLM achieves a better balance by maintaining multimodal understanding while addressing these inference challenges, offering both efficiency and performance.

## 6. Conclusion

We introduce Latent Language Modeling (LatentLM) for unified multimodal generation and understanding that seamlessly integrates continuous and discrete data using causal Transformers. By leveraging next-token diffusion, LatentLM achieves competitive performance across image generation, text-to-speech synthesis, and multimodal large language models. The method is scalable and practical for real-world applications. Future work will delve into more advanced multimodal-native reasoning (e.g., self-reflection for automatically refining generated images or tracking search states via latent vectors), extend to long video generation with interleaved script creation, explore cross-modal transfer strategies that leverage high token reduction ratios for seamless integration of speech and text, and further investigate embodied AI for end-to-end robot action and planning in continuous spaces.

## Acknowledgements

This work was supported in part by National Natural Science Foundation of China under Grant No. 62272264 and National Key Research and Development Program of China under Grant No. 2020YFA0804503.

## Impact Statement

This paper advances the capabilities of multimodal generative models by introducing a unified framework for continuous and discrete data. Such progress can bring a range of potential benefits, e.g., improved accessibility in user interfaces (through natural text, speech, and visual interaction), efficiency gains from working with lighter-weight representations, and new opportunities in fields such as education, health care, and content creation. However, like many large-scale generative models, deploying such systems carries risks, including possible misuse to generate deceptive or harmful content, increased automation that could replace certain forms of human labor, elevated potential for bias to be learned from uncurated training data, and challenges in handling user privacy if the model inadvertently encodes or reproduces sensitive information. Careful consideration of data curation practices, safety measures for output filtering, and transparency about model capabilities are key to mitigating these risks. We believe that it is important for practitioners to incorporate governance and oversight mechanisms, ensuring the continued responsible and equitable development of multimodal technologies.

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

## A. Comparison of Paradigms

Table 6 compares different approaches based on their attention patterns, native compatibility with causal autoregressive large language models (MLLM-Native), support for efficient inference (KV cache), and their ability to be trained for unified understanding and generation. Our proposed method, LatentLM, successfully incorporates the desirable properties.

| Paradigm | Attention | MLLM-Native | KV Cache | Unified Understanding-Generation Training |
|---|---|---|---|---|
| Discrete Token (VQ-MLLM, LlamaGen) | Causal | ✓ Yes | ✓ Yes | ✓ Yes |
| Image-Level Diffusion (Transfusion) | Vision: Bidirectional Text: Causal | ✗ No | ✗ No | ✗ No (noisy input) |
| Masked Autoregressive (MAR) | Bidirectional | ✗ No | ✗ No | ✗ No (incompatible with causal LM) |
| **LatentLM (Ours)** | Causal | ✓ Yes | ✓ Yes | ✓ Yes |

*Table 6.* A comparative analysis of multimodal LLM paradigms.

## B. Fixed Variance vs. Variance Sampling in $\sigma$-VAE.

For controlled ablations, such as the tokenizer-variance analysis in Figure 5, we instead use fixed variance values to isolate the effect of latent variance. Table 7 compares fixed standard deviation with Gaussian variance sampling during tokenizer training. Gaussian variance sampling consistently improves end-to-end generation quality, especially at intermediate and late tokenizer training stages.

| Tokenizer | 100k | 150k | 400k | 400k w/ CFG | 650k | 800k | 1000k | 1000k w/ CFG |
|---|---|---|---|---|---|---|---|---|
| Std 0.75 | 29.04 | 27.02 | 23.92 | 2.82 | 17.18 | 14.71 | 13.74 | 2.34 |
| Std 0.75 + Gaussian | 24.18 | 21.70 | 18.83 | 2.78 | 16.50 | 12.40 | 11.91 | 2.19 |

*Table 7.* End-to-end image generation results with fixed variance and Gaussian variance sampling in $\sigma$-VAE tokenizer training. We report FID scores at different tokenizer training steps. Lower is better.

## C. Contrast with Masked Autoregressive (MAR)

MAR (Li et al., 2024) highlights its contribution as autoregressive modeling with **masked bidirectional** attention. As shown in the MAR paper's Table 1, its "autoregressive" baseline (causal attention; raster order) is treated as a baseline variant, which underperforms the proposed MAR (i.e., bidirectional attention; randomly masked) method. The key novelty of our method lies in the redesigned sigma-VAE, enabling more effective autoregressive inference. With this key difference, we show that causal-attention autoregressive modeling works very well, which is contradictory to the MAR's findings. We highlight the comparison results in Figure 5. Compared to MAR, our model supports KV cache and is fully compatible with multimodal large language models.

## D. Comparison between $\beta$-VAE and $\sigma$-VAE

$\beta$-VAE (Higgins et al., 2016) sets different $\beta$ values to learn disentangled factors. In comparison, the $\sigma$-VAE used in this work sets a fixed variance $\sigma$. The motivations and implementations are different. The VAE models used in previous diffusion models typically perform more like an autoencoder (AE), i.e., with collapsed $\sigma$. This leads to inference degradation when combining diffusion with autoregressive decoding. Our contribution lies in identifying and resolving this specific issue within the diffusion-AR framework using a simple yet effective $\sigma$-VAE configuration, without introducing new complexity.

# E. MLLM Evaluation on Language Benchmarks

We evaluate the multimodal large language models (MLLMs) trained in Section 4.2 on various language benchmarks.

| Method | ARC-C | ARC-E | HellaSwag | OBQA | PIQA | Winogrande | Avg |
|---|---|---|---|---|---|---|---|
| LatentLM | 33.99 | 64.86 | 58.30 | 37.60 | 74.86 | 58.96 | 54.76 |
| Transfusion | 32.36 | 64.86 | 58.11 | 35.80 | 73.88 | 57.62 | 53.77 |
| Discrete Token | 30.30 | 63.09 | 55.11 | 36.40 | 73.18 | 53.39 | 51.91 |

*Table 8.* Accuracy on language benchmarks. The MLLMs are presented in Section 4.2.

# F. Generation Examples

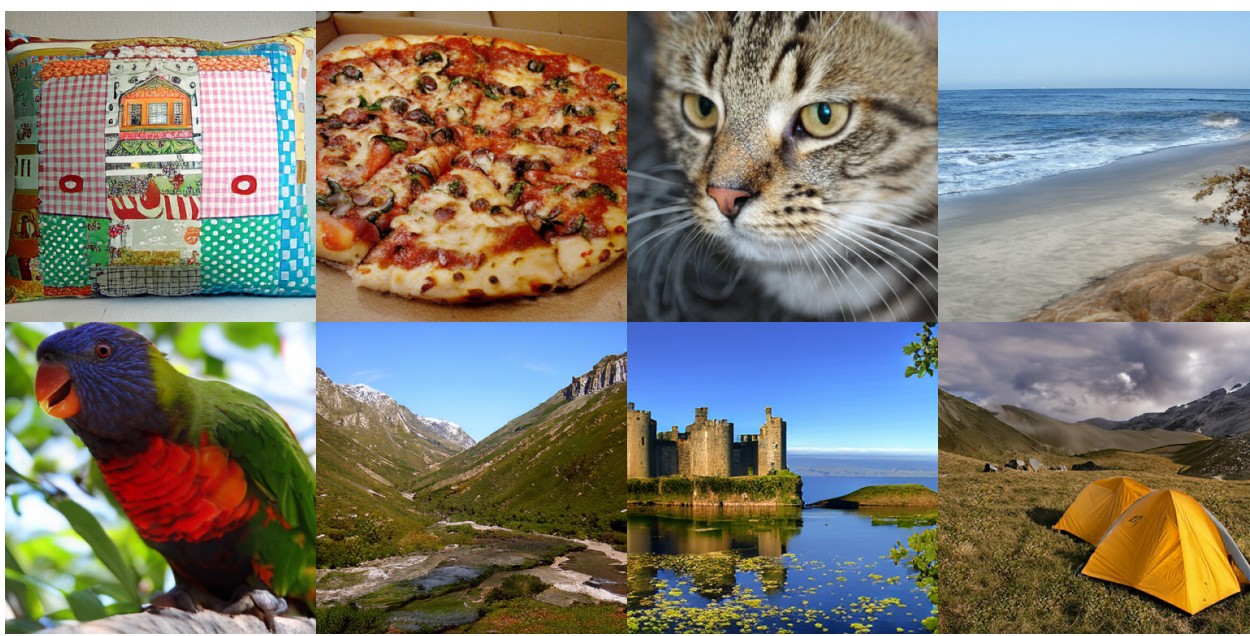

*Figure 7.* Samples of LatentLM trained on ImageNet. The resolution is 384×384. The images are generated by models described in Section 4.1.2.

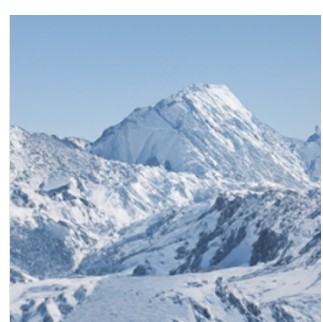 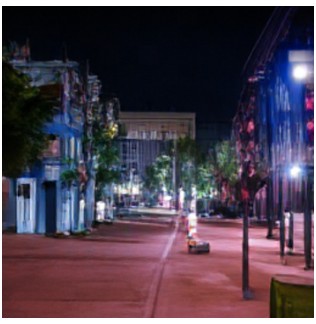 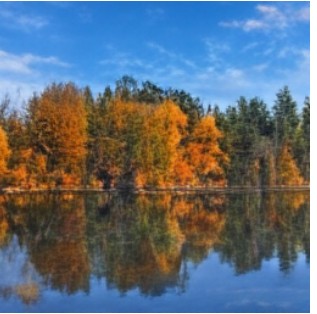 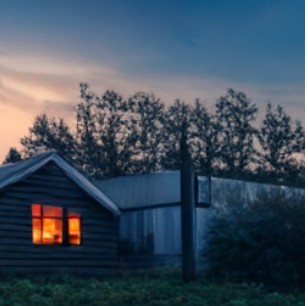

*Figure 8.* A majestic mountain range covered in snow.

*Figure 9.* A city street illuminated by lights.

*Figure 10.* A crystal lake surrounded by autumn trees.

*Figure 11.* A small house in a wooden at sunset.

*Figure 12.* Text-to-image examples of LatentLM. The images are generated by models described in Section 4.2.

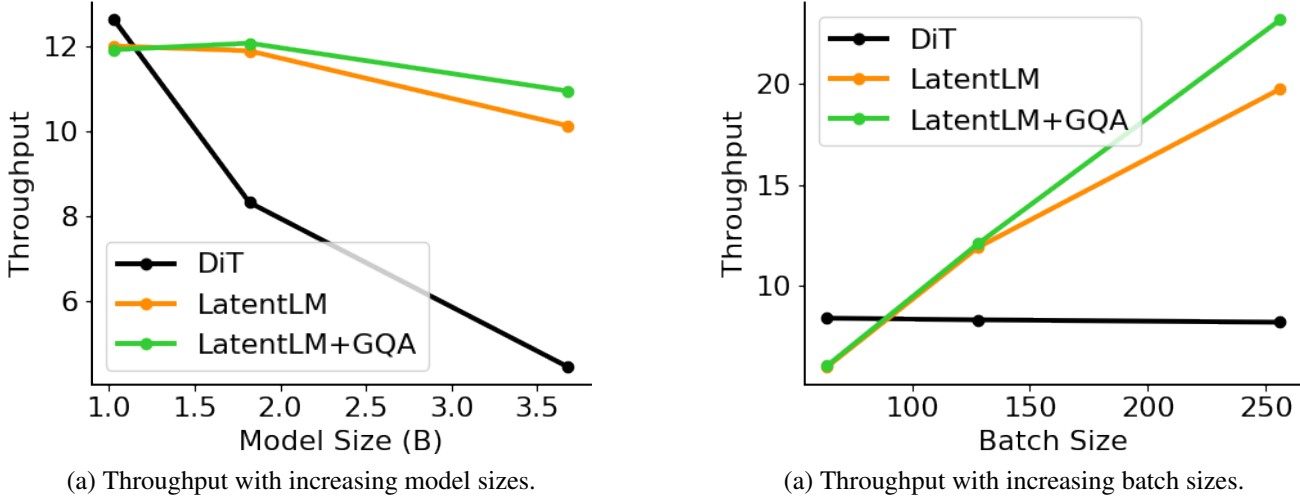

(a) Throughput with increasing model sizes.    (a) Throughput with increasing batch sizes.

*Figure 13.* We compare the inference throughput of Diffusion Transformer (Peebles & Xie, 2023) (DiT) and LatentLM in the settings of different model size and batch size. "GQA" stands for group-query attention (Ainslie et al., 2023).

## G. Inference Efficiency with Different Model Sizes

As shown in Figure 13, we investigate the inference capabilities of LatentLM by examining the effects of model size and batch size. We perform efficiency comparisons using 20 diffusion inference steps on a single H100 GPU.

First, we evaluate models ranging from 1B to 3.8B parameters with a fixed batch size of 128. Appendix G shows that DiT's throughput decreases significantly with larger model size. Because DiT has to iteratively perform multiple forward passes, it incurs higher computational costs. For the largest model with 3.8B parameters, LatentLM achieves a 2.47× increase in throughput, demonstrating its scalability advantages.

As presented in Appendix G, we then assess the 1.82B models with varying batch sizes. As the batch size increases, the throughput of LatentLM scales favorably with DiT. In addition, group-query attention (Ainslie et al., 2023) (GQA) further improves throughput. For a batch size of 256, our approach achieves a 2.84× throughput improvement. The results indicate that LatentLM benefits from significantly reduced FLOPs compared to image-level diffusion models, particularly at larger batch sizes.

As shown in Figure 14, we evaluate the efficiency with various model size and batch size. The results show that LatentLM's throughput increases with a larger batch size. Our approach benefits from key-value caches of causal Transformers, which avoids recomputation of history predictions. In contrast, DiT's throughput remains similar. In addition, group-query attention (Ainslie et al., 2023) (GQA) further improves the inference efficiency of LatentLM. Another advantage is that we can directly reuse the inference infrastructure of large language models to deploy LatentLM.

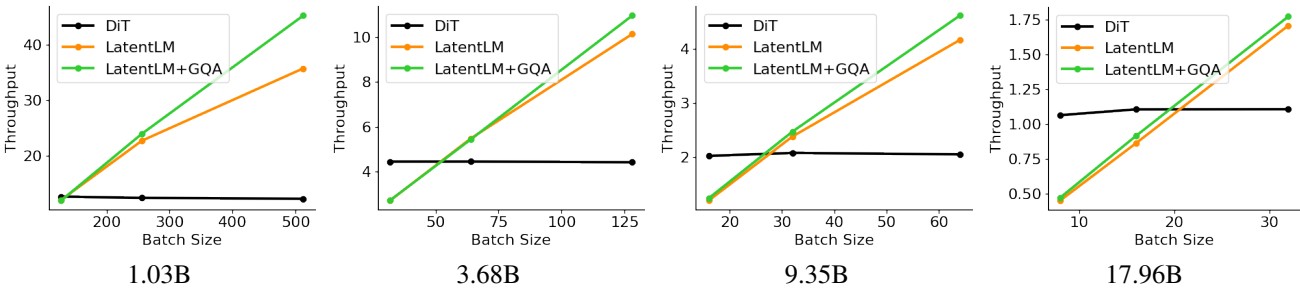

1.03B    3.68B    9.35B    17.96B

*Figure 14.* Inference throughput of various model size and batch size. "GQA" stands for group-query attention (Ainslie et al., 2023).

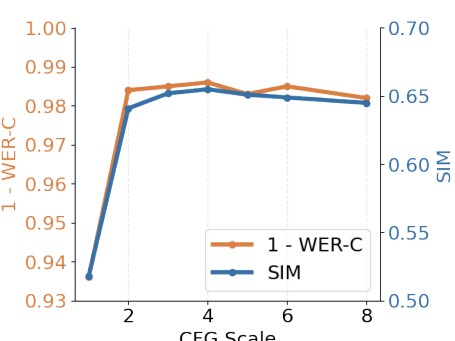 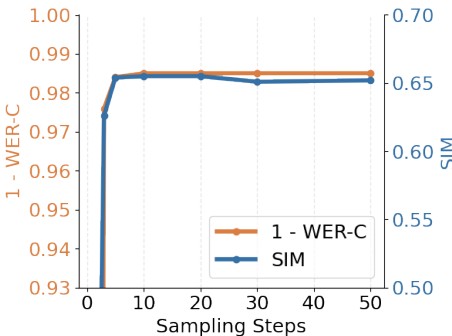

*Figure 15.* Ablation results of different CFG (Ho & Salimans, 2022) scales and inference sampling steps. We report zero-shot speech synthesis results.

## H. Ablation Studies of Text-to-Speech

**Token Reduction Ratio and Latent Dimension** We find that increasing the latent dimension enables the model to achieve a higher token reduction ratio and a lower frame rate. Table 9 presents the $\sigma$-VAE reconstruction and zero-shot speech synthesis results with different token reduction ratios and latent dimensions. We report the in-domain Mel distance performance of $\sigma$-VAE, along with the speaker similarity score and WER-C for tokenizer reconstruction and zero-shot speech generation on the LibriSpeech test-clean set. We use a 12-layer Transformer model for the TTS ablation studies. If the latent dimension remains unchanged, a higher token reduction ratio leads to a decrease in reconstruction performance and TTS speaker similarity score. However, by increasing the latent dimension of $\sigma$-VAE, we can compensate for this loss, allowing our model to use a higher token reduction ratio and a lower frame rate. Our model can generate 1 second of speech using significantly fewer autoregressive inference steps, compared to VALL-E 2.

**CFG Scale** Figure 15 illustrates the zero-shot speech synthesis results using classifier-free guidance (Ho & Salimans, 2022) (CFG). When the CFG scale is set to 1, CFG is not applied. The use of classifier-free guidance significantly enhances the model's performance. Furthermore, we find that setting the CFG scale to 4 yields the best results.

**Inference Sampling Step** Figure 15 presents the results of zero-shot speech synthesis using different inference sampling steps of the diffusion head. We set the CFG scale to 4 for the ablation studies. More sampling steps require more inference time. We find that a sampling step of 3 yields competitive results, and increasing it to 5 leads to further improvement. When the sampling step is increased further, the results improve only slightly. Using a sampling step of 5 allows the model to achieve strong performance while maintaining a fast inference speed.

**Human Evaluation for Text-to-Speech.** In addition to objective metrics, we conduct human evaluation for text-to-speech synthesis. The study involves 24 human annotators, with each annotator listening to approximately six hours of audio in total. As shown in Table 10, our model achieves the best average subjective score among the compared systems, while also obtaining strong objective WER and speaker similarity scores.

| Compression Ratio | Frame Rate | Latent Dimension | $\sigma$-VAE Reconstruction | | | Zero-Shot TTS | |
|---|---|---|---|---|---|---|---|
| | | | Mel Dist.↓ | SIM↑ | WER-C↓ | SIM↑ | WER-C↓ |
| 640× | 37.5 | 16 | 0.929 | 0.866 | 1.9 | 0.655 | 1.4 |
| 1600× | 15 | 16 | 1.080 | 0.700 | 2.7 | 0.545 | 1.6 |
| 1600× | 15 | 32 | 0.950 | 0.870 | 1.9 | 0.661 | 1.5 |

*Table 9.* Ablation results of different $\sigma$-VAE token reduction ratios and latent dimensions. We report tokenizer reconstruction quality and zero-shot speech synthesis.

| Model | Realism ↑ | Richness ↑ | Preference ↑ | Average ↑ | WER-W ↓ | WER-N ↓ | SIM-O ↑ |
|---|---|---|---|---|---|---|---|
| CosyVoice2 | – | – | – | – | 3.45 | 3.86 | 0.680 |
| MoonCast | – | – | – | – | 2.81 | 3.29 | 0.562 |
| SesameAILabs-CSM | $2.89 \pm 1.15$ | $3.03 \pm 1.11$ | $2.75 \pm 1.08$ | $2.89 \pm 1.12$ | 2.66 | 3.05 | 0.685 |
| Higgs Audio V2 | $2.95 \pm 1.13$ | $3.19 \pm 1.06$ | $2.83 \pm 1.16$ | $2.99 \pm 1.13$ | 5.94 | 5.97 | 0.543 |
| ElevenLabs v3 alpha | $3.34 \pm 1.11$ | $3.48 \pm 1.05$ | $3.38 \pm 1.12$ | $3.40 \pm 1.09$ | 2.39 | 2.47 | 0.623 |
| Gemini 2.5 Pro Preview TTS | $3.55 \pm 1.20$ | $3.78 \pm 1.11$ | $3.65 \pm 1.15$ | $3.66 \pm 1.16$ | 1.73 | 2.43 | – |
| Ours | $\mathbf{3.71 \pm 0.98}$ | $\mathbf{3.81 \pm 0.87}$ | $\mathbf{3.75 \pm 0.94}$ | $\mathbf{3.76 \pm 0.93}$ | **1.29** | **1.95** | **0.692** |

*Table 10.* Human evaluation and objective metrics for text-to-speech synthesis. Subjective scores are reported as mean ± standard deviation. Higher is better for Realism, Richness, Preference, Average, and SIM-O. Lower is better for WER-W and WER-N.

**Inference Latency of the Diffusion Head.** Although the diffusion head requires multiple denoising steps for each continuous token, it is lightweight in practice. The head is implemented as a small MLP without attention, so its per-token cost is nearly constant with respect to sequence length. Table 11 reports end-to-end TTS latency on a single NVIDIA A6000 with batch size 1. Increasing the number of diffusion steps from 1 to 10 increases the diffusion-head latency, but the overall real-time factor remains practical. This indicates that the diffusion head introduces noticeable but not dominant inference overhead.

| Model | Size | Diff. steps | LLM | Diff. head | Acoustic dec. | Sem. enc. | RTF ↓ |
|---|---|---|---|---|---|---|---|
| MoonCast | 1.5B | – | – | – | – | – | 1.43 |
| Higgs Audio V2 | 3B | – | – | – | – | – | 0.72 |
| Ours | 1.5B | 1 | 47.87 | 2.69 | 14.96 | 14.71 | 0.62 |
| Ours | 1.5B | 10 | 53.80 | 22.76 | 14.89 | 14.59 | 0.83 |

*Table 11.* End-to-end inference latency for text-to-speech synthesis. Latency is measured on a single NVIDIA A6000 with batch size 1. Component latency is reported in milliseconds. RTF denotes real-time factor.

# I. Experiment Settings

## I.1. Image Generation Scaling

Table 12 details the hyperparameters used for Section 4.1.2, where we compare the scalability properties of Diffusion Transformer (Peebles & Xie, 2023) (DiT) and LatentLM. We describe the hidden dimension, the number of layers, and the number of heads for the models. Specifically, we follow (Peebles & Xie, 2023) for the DiT architecture. In addition, we augment DiT with RMSNorm (Zhang & Sennrich, 2019) and SwiGLU (Ramachandran et al., 2017; Shazeer, 2020). To align the number of parameters, the FFN size for DiT is set to $\frac{8}{3}d$, while for LatentLM, it is set to $4d$. We train the models for 75,000 steps, which corresponds to approximately 120 epochs, to facilitate scaling comparisons.

| | Size | Hidden Dim. | #Layers | #Heads | Learning Rate |
|---|---|---|---|---|---|
| Medium | 455M | 1024 | 24 | 16 | $8 \times 10^{-4}$ |
| Large | 1.03B | 1536 | 24 | 12 | $3 \times 10^{-4}$ |
| XL | 1.82B | 2048 | 24 | 16 | $2 \times 10^{-4}$ |
| 3B | 3.68B | 2560 | 32 | 20 | $1.6 \times 10^{-4}$ |

*Table 12.* Model size and hyperparameters used for the scaling experiments in Section 4.1.2.

## I.2. Tokenizer Used for Image Generation

We train $\sigma$-VAE with perceptual loss (Zhang et al., 2018; Johnson et al., 2016) and GAN loss (Isola et al., 2017), following (Rombach et al., 2022; Esser et al., 2021). We initialize the encoder from the base-size BEiT-3 (Wang et al., 2023b)

checkpoint, and append a randomly initialized decoder. Both encoder and decoder have 12 Transformer layers, totaling 172 million parameters. The image patch size is 16. We train tokenizers on the ImageNet training set (Deng et al., 2009) with 200 epochs. The batch size is 256. The optimizer is AdamW (Loshchilov & Hutter, 2019) with $\beta = (0.0, 0.99)$ and a learning rate of 3e-4. The weight decay is set to 0.01. We apply layer-wise learning rate decay (Bao et al., 2022) of 0.65 on the encoder.

### I.3. Hyperparameters for Tokenizer Analysis

We follow the training recipes of (Peebles & Xie, 2023) for DiT and LatentLM training. We set the hidden size to 1024. The number of layers is 24. Because LatentLM does not have AdaLN in the Transformer backbone, we adjust the intermediate FFN dimension (i.e., 2730 in DiT, and 4096 in LatentLM) to match their model size. The diffusion head has three layers of feedforward networks.

We use the AdamW (Loshchilov & Hutter, 2019) optimizer with $\beta = (0.9, 0.98)$. We use the cosine learning rate schedule with a maximal value of 1e-4 and 100 warmup steps. The weight decay is 0.1. We train models using a batch size of 256 for 200,000 steps, which is approximately equivalent to 40 epochs. We use the cosine beta schedule and v-prediction (Salimans & Ho, 2022) for diffusion. We use DDPM (Ho et al., 2020) with 1000 steps during training. DPM-Solver (Lu et al., 2022a;b) with 20 steps is used during inference.

### I.4. Multimodal Large Language Model

**Training Data**   We use three types of data in the training stage: text-only data, image-text pair data, and interleaved text-image data. The mix-up ratio is 2:1:1. The data sources are described as follows:

- **Text-Only Data** The text training corpus follows (Sun et al., 2024b), including Common Crawl, RefinedWeb (Penedo et al., 2023), and StarCoder (Li et al., 2023).

- **Image-Text Pairs** We follow (Huang et al., 2023; Peng et al., 2023) to construct the paired data, i.e., English LAION-2B (Schuhmann et al., 2022), LAION-400M (Schuhmann et al., 2021), COYO-700M (Byeon et al., 2022), and Conceptual Captions (Sharma et al., 2018; Changpinyo et al., 2021).

- **Interleaved Image-Text Data** We use the same interleaved multimodal documents as in (Huang et al., 2023; Peng et al., 2023). The web pages are filtered from Common Crawl archives. The documents are interleaved with text and image.

**Configuration**   We train a 1.3B-size Transformer as the backbone. We set the hidden size to 2048. The number of layers is 24. The training sequence length is 4096. We use `tiktoken-cl100k_base` as the text tokenizer. The batch size is 4M tokens. We use the AdamW (Loshchilov & Hutter, 2019) optimizer with $\beta = (0.9, 0.98)$. The maximal learning rate is 3e-4 with 500 warmup steps. The total schedule is set to 1T tokens. We train the model with 50k steps (i.e., 200B tokens) for comparison.

Table 13 details the hyperparameters employed for multimodal large language models, as described in Section 4.2.

### I.5. Text-to-Speech Synthesis

#### I.5.1. TRAINING SETUP

Considering the variable-length nature of speech data, our speech tokenizer employs a convolutional architecture that supports streaming encoding and decoding. Specifically, $\sigma$-VAE for speech consists of a convolutional encoder, a continuous VAE quantizer, and a convolutional decoder. The encoder comprises multiple stages and downsampling layers organized in a hierarchical structure. Each stage includes several ConvNeXt blocks (Liu et al., 2022), where the original 2D convolution is replaced with 1D causal convolution. For token reduction ratios of 1600, 3200, and 6400, the downsampling layer reduces the input waveform by factors of [2, 4, 5, 5, 8], [4, 4, 5, 5, 8], and [4, 5, 5, 8, 8], respectively. Each time the downsampling layer is applied, the number of channels doubles, starting from 32 and increasing to 1024. The encoder contains around 120 million parameters in total. The decoder is a mirror of the encoder. As for the discriminator, we use the multi-period discriminator (Kong et al., 2020) and the complex STFT discriminator in DAC (Kumar et al., 2023).

The hidden size of LatentLM is 1024, with 24 layers and 16 attention heads. The intermediate FFN dimension is set to 4096. The diffusion head contains three layers of feedforward networks. We use the same Transformer architecture as VALL-E

| Params | Values |
| --- | --- |
| Layers | 24 |
| Hidden size | 2048 |
| FFN size | 6144 |
| Vocab size | 100,288 |
| Heads | 16 |
| Adam $\beta$ | (0.9, 0.98) |
| LR | $3 \times 10^{-4}$ |
| Batch size | 4M |
| Warmup steps | 500 |
| Weight decay | 0.1 |
| Head Layers | 6 |

*Table 13.* Hyperparameters used for multimodal large language models in Section 4.2.

2 (Chen et al., 2024) for comparison. Table 14 lists the hyperparameters utilized for text-to-speech synthesis models, as discussed in Section 4.3.

| Params | Values |
| --- | --- |
| Layers | 24 |
| Hidden size | 1024 |
| FFN size | 4096 |
| Heads | 16 |
| Adam $\beta$ | (0.9, 0.98) |
| LR | $7.5 \times 10^{-4}$ |
| LR schedule | cosine |
| Batch size | 5M |
| Warmup steps | 10k |
| Training steps | 100k |
| Weight decay | 0.01 |
| Head Layers | 3 |

*Table 14.* Hyperparameters used for text-to-speech synthesis in Section 4.3.

### I.5.2. TRAINING DATA

**Tokenizer** We train $\sigma$-VAE on a large and diverse corpus that includes speech, audio, and music. For speech, we use the clean speech subset from DNS Challenge 4 (Dubey et al., 2022) and all splits from the Common Voice v7 dataset (Ardila et al., 2020). For audio, we use the FSD50K dataset (Fonseca et al., 2021), along with the balanced and unbalanced splits from AudioSet (Gemmeke et al., 2017). For music, we use the MUSDB dataset (Rafii et al., 2017) and the Jamendo dataset (Bogdanov et al., 2019). All the data are resampled to 24kHz monophonic format.

**TTS Model** We utilize Libriheavy corpus (Kang et al., 2024) as training data following VALL-E 2 (Chen et al., 2024). This corpus is a labeled version of the Librilight corpus (Kahn et al., 2020), which features 50,000 hours of speech from approximately 7,000 different speakers, sourced from open-access English audiobooks associated with the LibriVox project[1].

---

[1] https://librivox.org

### I.5.3. EVALUATION METRICS

We evaluate our speech tokenizer using several automatic metrics, including: **Mel Distance**, which measures the distance between log Mel spectrograms as configured in DAC (Kumar et al., 2023); **PESQ-WB** (Rix et al., 2001), an intrusive metric for speech quality by comparing perceptual differences; **STOI** (Taal et al., 2010), which assesses speech intelligibility through short-time segment correlation; **VISQOL** (Chinen et al., 2020), a perceptual quality metric based on spectral similarity; **UTMOS** (Saeki et al., 2022), a reference-free mean opinion score for audio quality; **Speaker Similarity** (**SIM**), measured using WavLM-TDNN (Chen et al., 2022); and **Word Error Rate** (**WER**), calculated using both Conformer-Transducer (Gulati et al., 2020) (WER-C) and HuBERT-Large (Hsu et al., 2021) (WER-H) models.

