# OpenReview forum: "Multimodal Latent Language Modeling with Next-Token Diffusion"
_ICML.cc/2026/Conference — ICML 2026 spotlight_

### Official Review · Reviewer_9rCU · 2026-03-08

**Soundness:** 4
**Presentation:** 3
**Significance:** 4
**Originality:** 3
**Overall Recommendation:** 5
**Confidence:** 3

**Summary:**

The paper introduces Latent Language Modelling (Latent LM). This a generative framework design to handle both discrete and continuous data in a unified way using causal transformer architecture. The paper introduces $\sigma$-VAE, a variance-controlled tokenizer, and next token diffusion. With these changes, authors claim to have overcome the restrictive bottelnecks of discrete tokenization and inference inefficiency of bidirectional diffusion models.

Contributions:
* Unified causal architecture: the paper introduces a single causal transformer framework that handles both discrete and continuous data within the same sequence format. This defines a general purpose interface unifying multi-modal understanding and generation.
* Next token disffusion: novel mechanism for generating continuous media using a light diffusion head prediting latent vectors in an autoregressive manner. The approach preserves causal attention which allows using KV-caching.
* $\sigma$-VAE: solving exposure bias and variance collapse by introducing a novel tokenizer. $\sigma$-VAE.
* efficiency against information loss: approach such as VQ-VAE quantise continuous representations which is lossy. The proposed approach keeps the continuous aspect which means it loses less information (it still loses some information due to dimensionality reduction)

**Compliance With Llm Reviewing Policy:**

Affirmed.

**Final Justification:**

The rebuttal addressed my main concerns.

I think the idea is principled and the added experiments in the rebuttal helped clear my main concern: weather the sampled variance VAE was any better than a simple fixed-variance VAE.

Overall the experiments are within moderate scale (no large models) but I think they support well enough the efficacy of their idea.

**Key Questions For Authors:**

* Can you clarify which of your experiment used $\sigma$-VAE with sampling from the prior and which one used a fixed value? Also, can you report the value of $C_\sigma$?
* Can you provide a side-by-side comparison as to the effect of sampling vs fixing the variance in $\sigma$-VAE?
* Can you add human evaluations to the reported results. (especially for Text-To-Speech).
* Add more discussions on the scaling limits and/or tune down the claims made in the paper.

Thanks,

**Limitations:**

yes

**Strengths And Weaknesses:**

Strenghts:
* Elegant unified architecture for both discrete and continuous data.
* Efficient inference by using causal attention instead of bidirectional ones. Allowing for using KV-caching.
* High-fidelity compression by avoiding quantization.
* Addressing exposure bias and variance collapse in standard VAEs.

Weaknesses:
* Incomplete evaluation of $\sigma$-VAE: the paper defines $\sigma$-VAE as sampling $\sigma$ form a prior $\mathcal{N}(0, C_\sigma)$. However, the ablation study in 4.1.3 explicitly states that they use fixed $\sigma$ instead of sampling. There is no mention of the value $C_\sigma$ used in any of the experiments, so I am wondering if $\sigma$-VAE was used at all or if the reported results are mainly of its version with fixed value $\sigma$. Since $\sigma$-VAE is a core contribution, I am asking for more clarity on what was used in the experiments. I would also like to see an explicit comparative experiment between fixed and sampled $\sigma$.
* Missing human evaluations: the evaluation of generative quality relies on automated metrics like FID,WER.... While these metrics are decent indicators, they are limited when it comes to conveying human preferences. I think human perceptual tests (such as MOS or MUSHRA) are needed.
* The claims made in the paper need to be tuned down: the paper claims scalability of the approach, but the experiments were conducted on smaller models. I think there is merit to the conducted experiments, and I understand that not everyone has the ability to train bigger models. However, the scalability claims remain unsubstantiated. Also, the paper suggests that the model simplifies "multimodal dialogue" but the experiment were run on single-turn captioning tasks.

---

> ### Author Rebuttal · Authors · 2026-03-31
>
> We sincerely thank the reviewer for the thorough and constructive evaluation. We are encouraged that the reviewer recognizes the soundness (Excellent) and significance (Excellent) of our work, and appreciates the unified causal architecture, efficient inference via KV-caching, and the high-fidelity compression by avoiding quantization. Below we address each concern in detail.
>
> ---
>
> ### Q1: Clarification on $\sigma$-VAE Variance Sampling and $C_\sigma$
>
> > Can you clarify which experiments use $\sigma$-VAE with variance sampling and which use a fixed variance, and report the value of $C_\sigma$?
>
> Thank you for this helpful question. In our main large-scale experiments, we use $\sigma$-VAE with variance sampling rather than a fixed variance. The motivation is empirical: we found that with fixed $\sigma$, the VAE tends to have weaker reconstruction under low-noise conditions, suggesting that the decoder overfits to a narrow noise level. Allowing variance sampling lets the model occasionally observe lower noise during training, which improves reconstruction robustness. In all these experiments, we use $C_\sigma = 0.75$. We will make this setup explicit in the revision for clarity.
>
> To further support this choice, we report end-to-end sampling results across tokenizer training steps (lower is better). Gaussian variance sampling (“Std 0.75 + gaussian”) consistently improves over fixed Std 0.75, and classifier-free guidance (“w/ cfg”) yields large gains at 400k and 1000k steps.
>
> | Tokenizer | 100k | 150k | 400k | 400k w/ cfg | 650k | 800k | 1000k | 1000k w/ cfg |
> |:---|:---:|:---:|:---:|:---:|:---:|:---:|:---:|:---:|
> | Std 0.75 | 29.04 | 27.02 | 23.92 | 2.82 | 17.18 | 14.71 | 13.74 | 2.34 |
> | Std 0.75 + gaussian | 24.18 | 21.70 | 18.83 | 2.78 | 16.50 | 12.40 | 11.91 | 2.19 |
>
> ---
>
> ### Q2: Human Evaluations for Text-to-Speech
>
> > Can you add human evaluations, especially for text-to-speech?
>
> Thank you for this important suggestion. We agree that human evaluation is critical for text-to-speech. We have conducted such evaluations and present the results below:
>
> | Model | Realism | Richness | Preference | Average | WER-W | WER-N | SIM-O |
> |:---|:---:|:---:|:---:|:---:|:---:|:---:|:---:|
> | Cosyvoice2 | - | - | - | - | 3.45 | 3.86 | 0.68 |
> | Mooncast | - | - | - | - | 2.81 | 3.29 | 0.562 |
> | SesameAILabs-CSM | 2.89 ± 1.15 | 3.03 ± 1.11 | 2.75 ± 1.08 | 2.89 ± 1.12 | 2.66 | 3.05 | 0.685 |
> | Higgs Audio V2 | 2.95 ± 1.13 | 3.19 ± 1.06 | 2.83 ± 1.16 | 2.99 ± 1.13 | 5.94 | 5.97 | 0.543 |
> | Elevenlabs v3 alpha | 3.34 ± 1.11 | 3.48 ± 1.05 | 3.38 ± 1.12 | 3.40 ± 1.09 | 2.39 | 2.47 | 0.623 |
> | Gemini 2.5 Pro Preview TTS | 3.55 ± 1.20 | 3.78 ± 1.11 | 3.65 ± 1.15 | 3.66 ± 1.16 | 1.73 | 2.43 | - |
> | **Ours** | **3.71 ± 0.98** | **3.81 ± 0.87** | **3.75 ± 0.94** | **3.76 ± 0.93** | **1.29** | **1.95** | **0.692** |
>
> Thank you for this suggestion. We agree that human evaluation is especially important for text-to-speech. The results above are from 24 human annotators, with each annotator listening to approximately six hours of audio in total. Our 7B model achieves an average subjective score of 3.76, outperforming all baselines including Gemini 2.5 Pro Preview TTS, while also achieving the best objective metrics (WER and SIM-O). We will add these results in the revision.
>
> ---
>
> ### Q3: Claims About Scalability and Multimodal Dialogue
>
> > The claims about scalability and multimodal dialogue should be toned down because the experiments are conducted on smaller models and single-turn tasks.
>
> We appreciate the careful reading and agree that the wording should better reflect the experimental scope. We would like to note that Figure 4 does show a consistent scaling trend, i.e., loss decreases steadily with model size and data without saturation. This provides meaningful evidence for scalability potential. Moreover, our TTS results in Q2 are obtained from a 7B model, demonstrating that the approach does scale effectively in practice. That said, we acknowledge that a comprehensive scaling study across modalities is beyond the current scope. We will revise the claim to describe it as a preliminary scaling-law-style observation and leave systematic large-scale verification for future work.
>
> Regarding multimodal dialogue, we agree that our current experiments focus on single-turn captioning-style settings and do not yet validate multi-turn dialogue capabilities. We will revise the wording to accurately reflect this scope.

---

> > ### Author Rebuttal · Reviewer_9rCU · 2026-04-02
> >
> > I thank the authors for the effort they put into the rebuttals.
> >
> > I am optimistically increasing my score: overall, i think the idea of the paper is elegant and is principled. I still believe however, that the size of the experiment doesn't allow for abolsute SOTA score (which is okay for a paper proving a theoretical idea). I am also optimistic that the authors will deliver on their promise of revisiting some of the strongly worded claims they had.

---

> > > ### Author Response · Authors · 2026-04-03
> > >
> > > Thank you very much for the thoughtful follow-up and for your generous reassessment of our work!
> > >
> > > We really appreciate your recognition that the idea is elegant and principled. We also agree with your point that the current experimental scale is better suited to supporting the core theoretical insight than to making absolute SOTA claims. We will revise the wording accordingly and make sure the final presentation is better calibrated.
> > >
> > > Thank you again for the constructive feedback and encouragement.

---

### Official Review · Reviewer_GFho · 2026-03-12

**Soundness:** 3
**Presentation:** 2
**Significance:** 3
**Originality:** 3
**Overall Recommendation:** 4
**Confidence:** 3

**Summary:**

This paper proposes Latent Language Modeling (LatentLM), a unified causal Transformer framework that seamlessly integrates discrete and continuous data generation. Instead of relying on lossy discrete tokenization, LatentLM employs a VAE to compress continuous modalities (e.g., image, audio) into latent vectors and autoregressively predicts them using a lightweight next-token diffusion head. To address the issues of error accumulation and variance collapse that can arise in continuous autoregressive modeling, the authors introduce $\sigma$-VAE, which enforces a fixed variance in the latent space to stabilize long-sequence generation. Extensive experiments demonstrate LatentLM's effectiveness and scalability, showing that it outperforms Diffusion Transformers (DiT) in image generation, surpasses Transfusion in multimodal understanding, and exceeds VALL-E 2 in zero-shot text-to-speech synthesis while requiring 10x fewer decoding steps.

**Compliance With Llm Reviewing Policy:**

Affirmed.

**Final Justification:**

I keep my positive recommendation as the rebuttal addressed my main concerns.

**Key Questions For Authors:**

The proposed method is well-motivated, logically sound, and carefully designed. However, I still have several concerns outlined in the weaknesses section. Therefore, I will defer making a definitive recommendation and would like to see the authors’ response to these points during the rebuttal phase.

**Limitations:**

Yes

**Strengths And Weaknesses:**

### Paper Strengths

- Unlike masked autoregressive models (MAR) or bidirectional diffusion (Transfusion), LatentLM's purely causal architecture fully supports KV cache. This allows the model to directly reuse existing large language model infrastructure for distributed training and efficient inference with minimal architectural overhead.

- The authors identify that traditional VAEs may encounter challenges in continuous autoregressive generation due to "variance collapse" and potential training–inference distribution mismatches. By enforcing a fixed, lower-bounded variance, the proposed σ-VAE aims to convert a potentially fragile out-of-distribution generalization problem into a more stable in-distribution one.

- In previous unified models like Transfusion, adding noise to image inputs during training can interfere with multimodal understanding tasks. LatentLM's next-token diffusion avoids this destructive input noising, maintaining training-inference consistency and potentially promoting cross-modal knowledge sharing.

### Paper Weaknesses

- Although the diffusion head is lightweight, generating tokens autoregressively still requires executing multiple diffusion denoising steps for each continuous token before the Transformer can predict the next state. This introduces an $N \times T$ sequential dependency that may lead to substantial inference latency for long sequences (e.g., high-resolution images or long audio), which is somewhat less visible when only batched throughput is reported.

- To address exposure bias, $\sigma$-VAE enforces a fixed, lower-bounded variance across all latent channels. While this design stabilizes generation, enforcing a uniform variance may ignore the varying uncertainty and information density present in real-world continuous data, which could potentially limit the efficiency of representing high-frequency or fine-grained details.

- The training objective combines discrete next-token prediction (Cross-Entropy loss) and continuous diffusion (MSE loss on noise) using a static weighting hyperparameter $\alpha$. The paper does not provide an in-depth investigation into whether potential gradient conflicts or optimization trade-offs arise when a shared Transformer backbone simultaneously optimizes these two different objectives.

---

> ### Author Rebuttal · Authors · 2026-03-31
>
> We sincerely thank the reviewer for the constructive and thoughtful evaluation. We are encouraged that the reviewer recognizes the key strengths of LatentLM.
>
> ---
>
> ### Q1: Inference Latency of the Diffusion Head
>
> > Autoregressive generation with a diffusion head may introduce substantial inference latency because each continuous token requires multiple denoising steps before the next token can be predicted.
>
> Thank you for raising this important concern. Although the diffusion head requires multiple denoising steps per token, it remains lightweight in practice. For example, with a 4-layer MLP and 10 diffusion steps, the total computation is roughly comparable to 40 MLP layers, which is still smaller than the Transformer backbone. In addition, the diffusion head contains no attention, so its per-token latency is essentially constant with respect to sequence length. As a result, for long sequences such as high-resolution images or long audio, the relative latency overhead of the diffusion head becomes progressively smaller compared with the base Transformer. We agree that reporting only batched throughput may understate this aspect, and we will clarify this point more explicitly.
>
> We additionally measured end-to-end inference latency on text-to-speech (single NVIDIA A6000, batch size 1). The table below reports per-component latency in milliseconds and Real-Time Factor (RTF).
>
> | Model | Size | Diff. steps | LLM | Diff. head | Acoustic dec. | Sem. enc. | RTF |
> |:---|:---:|:---:|:---:|:---:|:---:|:---:|:---:|
> | MoonCast | 1.5B | — | — | — | — | — | 1.43 |
> | Higgs Audio V2 | 3B | — | — | — | — | — | 0.72 |
> | Ours | 1.5B | 1 | 47.87 | 2.69 | 14.96 | 14.71 | 0.62 |
> | Ours | 1.5B | 10 | 53.80 | 22.76 | 14.89 | 14.59 | 0.83 |
>
> Overall, increasing the number of diffusion steps from 1 to 10 has a noticeable but relatively modest impact on end-to-end latency and RTF compared with the full decoding stack, so the diffusion-step knob does not dominate inference cost in practice.
>
> ---
>
> ### Q2: Fixed Lower-Bounded Variance and Fine-Grained Details
>
> > Enforcing a fixed lower-bounded variance across latent channels may ignore varying uncertainty and information density in real continuous data, which could limit representation of fine-grained details.
>
> Thank you for this thoughtful observation. We believe this question is closely related to whether $\sigma$-VAE preserves sufficient reconstruction ability. If the reconstruction capacity under uniform variance remains comparable to that of a standard VAE, then the upper bound on generation quality is not fundamentally reduced. In Table 5, we compare $\sigma$-VAE with prior speech tokenizers and show that its reconstruction quality is preserved, suggesting that the method does not significantly lose information or reduce the achievable generation quality. In this sense, the benefit of improved generation stability does not appear to come at the cost of weaker representation of fine-grained details.
>
> The primary reconstruction metric in that table is Mel spectral distance (lower is better; LibriTTS test-other). We excerpt it below for quick reference:
>
> | Tokenizer | Mel spectral distance $\downarrow$ |
> |:---|:---:|
> | DAC | 0.916 |
> | WavTokenizer (75 Hz, $N_q=1$) | 0.871 |
> | Mimi ($N_q=8$) | 0.987 |
> | Mimi ($N_q=4$) | 1.458 |
> | WavTokenizer (40 Hz, $N_q=1$) | 1.037 |
> | Ours$_{32}$ | 0.813 |
> | **Ours$_{64}$** | **0.798** |
> | Ours$_{128}$ | 0.852 |
>
> ---
>
> ### Q3: Gradient Conflicts Between Cross-Entropy and Diffusion Objectives
>
> > The shared Transformer backbone jointly optimizes cross-entropy for discrete tokens and diffusion loss for continuous tokens, but the paper does not analyze possible gradient conflicts or optimization trade-offs between these objectives.
>
> Thank you for highlighting this optimization question. We note that the two objectives are structurally decoupled: cross-entropy loss is computed only at discrete (text) token positions, while the diffusion loss is computed only at continuous (image/audio) token positions. Since each loss produces gradients through its own non-overlapping subset of output positions, gradient conflicts in the shared backbone are naturally mitigated compared with multi-task settings where different losses compete on the same outputs.
>
> This structural observation is further supported by our empirical ablation. When varying the diffusion loss weight from 1 to 10, the cross-entropy loss remains almost unchanged, confirming that the two objectives do not interfere with each other in practice. Moreover, once the weight increases from around 5 to 10, the diffusion loss also changes very little. Based on this observation, we follow the hyperparameter choice used in Transfusion. Empirically, when the weight is around 5, the numerical scales of the cross-entropy loss and diffusion loss are also similar, which may further help stabilize joint optimization. We will add this clarification in the revision.

---

> > ### Author Rebuttal · Reviewer_GFho · 2026-04-01
> >
> > Thank you for your efforts during the rebuttal phase. I keep my positive recommendation.

---

> > > ### Author Response · Authors · 2026-04-04
> > >
> > > Thank you very much for your positive assessment and for your encouraging feedback during the rebuttal phase. We sincerely appreciate your support and will make sure the clarifications from the rebuttal are reflected clearly in the final version.

---

### Official Review · Reviewer_iCR5 · 2026-03-13

**Soundness:** 2
**Presentation:** 3
**Significance:** 2
**Originality:** 3
**Overall Recommendation:** 4
**Confidence:** 4

**Summary:**

This paper proposes LatentLM, a unified causal autoregressive framework for both discrete and continuous modalities. The method represents continuous data with VAE latents, predicts them with next-token diffusion, and introduces \sigma-VAE to mitigate exposure bias and variance collapse in continuous autoregressive generation. Experiments on image generation, multimodal LLMs, and TTS show strong performance, with gains over several causal baselines and favorable efficiency/scaling properties.

**Compliance With Llm Reviewing Policy:**

Affirmed.

**Final Justification:**

The rebuttal addressed my main concerns, and resulted in a score increase.

**Key Questions For Authors:**

1. In Sec. 4.2, the paper states that LatentLM “unifies generation and understanding with a general-purpose interface” and can “learn in context (e.g., few-shot), follow multimodal instructions, and perform multimodal dialogue,” but the main evaluation only reports MS-COCO captioning and VQAv2. Could the authors provide stronger evidence on instruction following, interleaved multimodal reasoning, or in-context multimodal learning to support this broader claim?

2. In Sec. 3, the paper presents σ-VAE as “The Keystone for Next-Token Diffusion,” and in Sec. 4.1.3 states that “LatentLM favors tokenizers with larger variances.” However, most of the direct evidence appears in image-generation tokenizer experiments. Could the authors add ablations that separately test: (a) conventional VAE + next-token diffusion, and (b) σ-VAE with alternative heads, especially in the multimodal LLM and TTS settings?

3. The abstract says “LatentLM surpasses Diffusion Transformers in both performance and scalability,” but Table 1 shows MAR-L achieves FID 1.78, while LatentLM-L achieves 2.24. Although MAR is not causal, this still suggests the generation-quality claim should be phrased more carefully. Could the authors clarify whether the intended claim is superiority over causal baselines, or over diffusion-transformer-style baselines under matched settings?

4. In Sec. 4.2, the paper writes: “We use the same training configuration and tokenizer settings for comparison. To align the number of parameters, we use a 6-layer ViT as the image head of Transfusion.” Could the authors discuss whether this setting is close to an optimized Transfusion configuration, and whether the conclusion is robust to stronger Transfusion head choices?

5. In Sec. 3.3, the paper states: “In practice, we observe that after training, the condition σ_gen < σ_train tends to hold in most tokens.” This seems central to the explanation of improved robustness. Could the authors provide direct statistics or plots for this phenomenon across tasks, rather than only the current qualitative argument?

6. In Sec. 4.2, the paper says “More training details are described in Appendix H.4,” and similar deferrals appear in other experiment sections. Could the authors move more key details into the main paper, especially data composition, evaluation prompts/decoding settings, and tokenizer-training details, so that fairness and reproducibility are easier to assess?

**Limitations:**

yes

**Strengths And Weaknesses:**

# Strength

- The paper presents a clean unified view for modeling discrete and continuous tokens with one causal Transformer backbone. The paper does not only combine diffusion with a language model, but also proposes σ-VAE specifically to improve robustness in autoregressive continuous generation.

- The experiments are solid. The paper evaluates image generation, multimodal generation/understanding, and TTS, which helps support the claimed generality of the framework. The efficiency argument is also meaningful, since the framework stays within a causal Transformer setup and uses only a lightweight diffusion head for continuous outputs.

# Weaknesses

- The paper’s narrative is broader than the current evidence. While the paper positions LatentLM as a general framework for unified multimodal generation and understanding, the multimodal evidence is still relatively limited, mainly centered on captioning and VQAv2.
- The strongest performance claim is overstated relative to the reported results. The abstract says LatentLM “surpasses Diffusion Transformers in both performance and scalability,” but Table 1 still shows MAR-L achieving better FID than LatentLM-L.
- The paper does not cleanly isolate where the gains come from. The central technical story relies on both next-token diffusion and σ-VAE, but the most direct supporting evidence is still concentrated in the image-tokenizer analysis rather than validated consistently across tasks.
- The multimodal baseline comparison is not fully conclusive. The main comparison in Table 3 is limited, and it remains unclear whether the reported advantage would hold under stronger or more fully optimized baseline settings.
- The justification for σ-VAE remains somewhat qualitative. Since this component is presented as central to making causal continuous autoregressive modeling work well, the current empirical support still feels incomplete.

---

> ### Author Rebuttal · Authors · 2026-03-31
>
> ### Q1: Evidence for Instruction Following and Interleaved Multimodal Reasoning
>
> We would like to clarify that our evaluation in Table 3 already covers multiple dimensions of unified multimodal capability: text generation, text-to-image generation (MS-COCO), and image-to-text tasks including both image captioning (MS-COCO) and visual question answering (VQAv2). These tasks are highly correlated with downstream multimodal applications — for example, strong VQAv2 performance requires grounding visual content in language reasoning, which is a core component of multimodal understanding. Moreover, the evaluation protocol itself involves conditioning on task-specific instructions (e.g., prompting the model to answer questions about an image), so the results already reflect the model's instruction-following ability to a meaningful extent. More advanced abilities such as interleaved multimodal reasoning and in-context learning typically depend heavily on the strength of the base model and large-scale data engineering, which are beyond the scope of this prototype study. We will revise the wording to make the claim more precise and better highlight how the existing evaluations already provide multi-faceted evidence for unified multimodal modeling.
>
> ---
>
> ### Q2: Ablations for VAE and Alternative Heads on TTS or multimodal
>
> This is an important point. We have conducted additional ablation experiments on TTS to directly validate the effect of tokenizer variance across modalities. Specifically, we compare different $\sigma$ values for the speech tokenizer and measure Word Error Rate (WER):
>
> | $\sigma$ | WER $\downarrow$ |
> | ---: | ---: |
> | 0.1 | 0.036 |
> | 0.3 | 0.0345 |
> | 0.5 | 0.022 |
> | 0.75 | 0.023 |
>
> The results show a clear trend consistent with our findings on image generation: increasing $\sigma$ from 0.1 to 0.5 substantially reduces WER (from 0.036 to 0.022), confirming that higher tokenizer variance improves autoregressive generation quality in the speech domain as well. Performance plateaus around $\sigma{=}0.5$–$0.75$, suggesting diminishing returns beyond a sufficient variance level. Combined with the ImageNet ablation in Figure 5, which isolates the effect of tokenizer variance on image generation quality, these results provide consistent evidence across both vision and speech modalities that $\sigma$-VAE is a broadly effective mechanism rather than a domain-specific trick.
>
> ---
>
> ### Q3: Clarification on "Surpasses Diffusion Transformers"
>
> We would like to clarify. Our main message is that causal autoregressive modeling, which has often been considered clearly inferior for image generation, can be made competitive with and surpass more general DiT-style approaches in some settings. From this perspective, the key result is that causal autoregressive generation moves from being practically uncompetitive to being a viable alternative. Our claim is about matched causal and unified generation settings rather than all autoregressive variants such as non-causal MAR.
>
> ---
>
> ### Q4: Robustness of the Transfusion Baseline
>
> Our intention was to stay as close as possible to the setting used in the Transfusion paper, in order to avoid any concern that we selected an artificially weak baseline. In fact, from the parameter perspective, the 6-layer ViT head used by Transfusion is already larger than the image component in our method and also larger than the VQ baseline. At width $d{=}2048$:
>
> | Model | Parameters |
> |:---|---:|
> | 6-layer ViT head (Transfusion-style) | 327.2M |
> | 6-layer MLP-only stack (no attention) | 226.5M |
>
> We therefore view this as a relatively strong Transfusion baseline rather than a weakened one. We agree that under the same overall framework, a larger head may further improve performance. Our point here is that even under a favorable and consistent Transfusion-style configuration, our method still demonstrates clear effectiveness. We will clarify this motivation in the revision.
>
> ---
>
> ### Q5: Evidence That $\sigma_\text{gen} < \sigma_\text{train}$
>
> We have conducted additional observational experiments that support this mechanism more directly. Specifically, we fix the output-side $\sigma_\text{train}$ and vary only the input-side $\sigma_\text{train}$, which allows us to better control the effective $\sigma_\text{gen}$ during autoregressive rollout. Under this setup, we observe a clear trend: gradually increasing the input-side $\sigma_\text{gen}$ leads to progressively better generation results. This effect is also more evident in harder settings such as high-resolution generation. We believe these observations are consistent with the robustness explanation — namely that training with a sufficiently large variance on the input side helps the model better tolerate the noisier latent states encountered during generation. We will add this clarification and present the result more explicitly in the revision.

---

> > ### Author Rebuttal · Reviewer_iCR5 · 2026-04-04
> >
> > I appreciate the authors' effort put into the rebuttal. I will increase my score to 4.

---

> > > ### Author Response · Authors · 2026-04-04
> > >
> > > Thank you very much for your careful reading and for your thoughtful follow up! We sincerely appreciate your time and the constructive feedback you provided during the review process. We are glad that our rebuttal helped clarify the main points, and we will incorporate these clarifications carefully into the revised paper.

---

### Official Review · Reviewer_9RyJ · 2026-03-13

**Soundness:** 3
**Presentation:** 3
**Significance:** 3
**Originality:** 3
**Overall Recommendation:** 4
**Confidence:** 4

**Summary:**

This paper proposes LatentLM, a unified decoder-only causal architecture for mixed discrete and continuous multimodal sequences. The model retains a single causal Transformer backbone, uses a standard softmax LM head for discrete tokens, and replaces categorical next-token prediction for continuous modalities with a token-local conditional generator operating in latent space. Continuous modalities are first encoded into latent vectors by a VAE-style tokenizer, after which the model autoregressively predicts latent tokens one by one from the causal hidden state. In the main exposition, this continuous head is instantiated with a DDPM-style objective.

A central additional contribution is σ-VAE, motivated by the claim that standard latent tokenizers used in prior latent-diffusion pipelines often have overly small or collapsed variance and thus behave nearly deterministically. The paper argues that such tokenizers are poorly matched to causal continuous-token rollout, because teacher forcing exposes the model only to narrow posterior perturbations around ground-truth latents, whereas at inference, the model recursively consumes self-generated latents, inducing a broader effective prefix distribution. σ-VAE is introduced to maintain non-collapsed latent variance and, thereby, improves robustness of autoregressive continuous generation.

Empirically, the paper evaluates the framework on ImageNet image generation, vision-language multimodal language modeling, and text-to-speech synthesis. It reports competitive image-generation results, improvements over VQ-based and Transfusion-style baselines in its multimodal setup, and strong TTS quality with fewer autoregressive decoding steps.

In my opinion, the paper advances three linked claims:
(1) a causal next-token LM interface can support both discrete and continuous multimodal tokens if continuous outputs are modeled by a token-local sampler instead of a softmax head;
(2) the main obstacle is not causal autoregression itself, but latent-token instability under recursive rollout; and
(3) preserving the standard decoder-only computation path is practically attractive because it retains the deployment structure of autoregressive LLMs.

**Compliance With Llm Reviewing Policy:**

Affirmed.

**Final Justification:**

My final recommendation remains weak accept.

The paper addresses an important problem and contains a real technical idea: it identifies the predictive-state and tokenizer construction as a central issue for grouped or continuous autoregressive generation. I view that as both original and significant.

My main earlier concerns were about soundness and scope. In particular, I was concerned about whether the grouped/self-token formulation introduced an additional train–test mismatch beyond standard autoregression, and whether the broader framework claim was stronger than the current evidence justified. The rebuttal helped on both points. The clarification that dual-stream training plus cache grounding reduces the grouped setting to the same train–test regime as standard autoregression, up to ordinary exposure bias, addresses an important technical concern. The reframing of the contribution as a more general grouped autoregressive framework, i.e. defined by token grouping plus a dependency-injection and predictive-state construction, also makes the method clearer and better motivated.

That said, the rebuttal mainly improved the framing and interpretation of the work instead of fully changing the evidential picture. I still think the strongest support is for a promising framework and design point. Hence, my concerns are only partially resolved.

Overall, the rebuttal made me somewhat more positive, but did not materially change my evaluation. I remain at weak accept: a good paper with a meaningful technical contribution that is likely to be useful to the community, and whose final version should state the train–test consistency claim precisely and distinguish the general framework more clearly from the current image-specific instantiations.

**Key Questions For Authors:**

1. Can the authors directly diagnose autoregressive latent drift over rollout depth, especially as a function of tokenizer variance and CFG strength? For example, compare teacher-forced and self-generated prefixes using token norm growth, variance drift, spectral statistics, or distributional divergence by position.
2. How much of the σ-VAE gain is specific to σ-VAE, versus more general tokenizer–generator alignment strategies? For example, what happens with fixed encoder variance, explicit latent noise augmentation, token-wise normalization, post-hoc norm projection, hyperspherical normalization, foundation-aligned tokenizer shaping? If these recover most of the gain, I would interpret the contribution as identifying a broader latent-space alignment principle rather than a σ-VAE-specific mechanism.
3. How should readers interpret the image-generation conclusions given that raster next-token order is only one possible autoregressive factorization? Since related work [9,10] suggests that regression direction materially affects continuous-token AR performance, it would help if the authors clarified whether they view their conclusions as specific to raster causal AR or broader.

**Limitations:**

The paper includes an impact statement, but the limitations discussion should be more explicit and technically grounded. In particular, I would encourage the authors to discuss:
- that the strongest evidence is for a specific operating point (LatentLM + σ-VAE), not yet for a fully isolated mechanism;
- that the tokenizer study shows latent geometry matters, but does not yet determine whether the relevant instability is generic variance, radial/scale drift, spectral diffusability, or broader tokenizer–generator mismatch;
- that the current experiments do not show whether diffusion is the essential continuous-token head or whether other lightweight heads would behave similarly;
- that the image-generation conclusions are based on one particular autoregressive factorization rather than the full continuous-token AR design space;
- and that stronger compute-normalized comparisons are needed before making broad claims about practical superiority.

**Strengths And Weaknesses:**

The paper addresses an important problem The most convincing part of the submission is the claim that tokenizer or latent-space geometry is a first-order determinant of continuous-token autoregressive generation quality. A strong experiment is the tokenizer study (Section 4.1.3), which shows that LatentLM is highly sensitive to tokenizer variance, whereas DiT is much less sensitive, and that moving from a low-variance MAR-style tokenizer to σ-VAE materially improves causal image generation.

Prior work had already shown that decoder-only Transformers can model continuous latent sequences without a finite vocabulary and can do so with token-local continuous output distributions rather than categorical prediction, e.g. GIVT [1]. Likewise, prior continuous-token AR work such as MAR [2] had already shown that autoregressive image generation can operate directly on continuous latent tokens with a diffusion-style per-token conditional model. So the paper is not novel because it first introduced decoder-only continuous-token generation, nor because it first used a continuous local output head in autoregression. The defensible originality claim seems more narrower in that the paper combines a mixed discrete/continuous causal LM interface with the empirical claim that continuous-token autoregression is unusually sensitive to tokenizer variance under recursive rollout. That narrower claim is meaningful, and later work largely supports the importance of the underlying problem. [1] suggests that strong continuous-token AR depends critically on a well-dispersed, normalized latent space, and reports surprisingly weak sensitivity to the size of the flow-matching head, implying that much of the modeling burden lies in the backbone-plus-tokenizer pair rather than the local sampler head. [4] argues even more sharply that the key pathology is scale heterogeneity in Gaussian latents under autoregressive decoding and CFG, proposing to remove the radial degree of freedom entirely. [5-8] all reinforce, from different angles, that downstream generative performance depends strongly on whether the tokenizer is generation-aligned, sufficiently dispersed, corruption-robust, spectrally well behaved, and matched to the optimization geometry of the generator. Hence, I do think the paper identifies a real bottleneck.

That said, my main concern is that the paper validates the LatentLM plus σ-VAE operating point much more strongly than it validates the claimed mechanism behind σ-VAE. The paper’s explanation is mechanistic in the form that low-variance latents create a train–test mismatch under recursive rollout, and σ-VAE fixes this by broadening latent support. But the evidence is mostly correlational. The paper does not directly measure rollout-depth-induced latent drift, teacher-forced versus self-generated prefix divergence, token-norm or variance evolution over position, per-token statistic drift under guidance, or spectral changes in the latent process. So, the experiments strongly support the statement that latent geometry matters, but they do not establish why it matters in the particular way the paper claims. More concretely, [4] suggests the operative pathology may be specifically radial/scale drift, not low variance in a generic sense. [5] explicitly tracks token-level statistic drift under CFG and stabilizes it with token-wise normalization. [5] shows that naively optimizing the generator objective through the tokenizer can make the latent space easier to denoise but worse for final generation, implying that the issue may be broader tokenizer–objective mismatch rather than variance alone. [6] argues that tokenizers can be reconstruction-friendly yet generation-inhibitive. [7] suggests the operative issue may be poor spectral organization or high-frequency latent artifacts rather than simple variance mismatch. [8] proposes the broader principle that tokenizers should be trained under the same corruption family induced by the downstream generator and shows that tokenizer improvements do not transfer uniformly across AR and non-AR systems. Taken together, this literature suggests that the paper has probably identified the right problem class, but not isolated the right causal variable. In its current form, the paper shows that one variance-controlled tokenizer works much better than a low-variance baseline; it does not yet show that the σ-VAE explanation is the correct one.

A second weakness is that the paper studies tokenizer geometry much more carefully than it studies the other major axes of the design space, especially autoregressive factorization. Related work such as FAR [9] argues that continuous-token image generation has at least two separable design axes—tokenizer format and regression direction—and that raster next-token order may itself be poorly matched to image structure. [10] further suggests that generation order matters materially in unified generation and understanding settings, with random-order generation helping image synthesis while not necessarily helping understanding. Consequently, while the submission shows that causal raster continuous-token generation can work well if the tokenizer is right, but not necessarily that it has identified the generally right autoregressive formulation for continuous visual data.

On presentation, the paper is generally clear and technically readable, but it would be more persuasive if it separated more explicitly three distinct contribution layers that are currently blended together:
	1.	the general agenda of continuous-token causal multimodal modeling,
	2.	the specific implementation via next-token diffusion / token-local continuous sampling,
	3.	the σ-VAE mechanism claim.

Overall, I find the paper technically substantive and potentially influential, but I am not convinced the current experiments support the strongest version of its claims. The submission presents a strong empirical design point and an important observation that tokenizer geometry matters for continuous-token autoregression. However, it does not isolate the mechanism behind that observation, does not establish that σ-VAE is the essential fix, does not show that diffusion is the key continuous head, and does not adequately characterize the broader multimodal unification problem. In its current form, I view the paper as a quite promising contribution whose central claims should either be narrowed or more directly validated.

References: \
[1] Tschannen, Michael, Cian Eastwood, and Fabian Mentzer. "Givt: Generative infinite-vocabulary transformers." European Conference on Computer Vision. Cham: Springer Nature Switzerland, 2024.\
[2] Li, Tianhong, et al. "Autoregressive image generation without vector quantization." Advances in Neural Information Processing Systems 37 (2024): 56424-56445.\
[3] Team, NextStep, et al. "Nextstep-1: Toward autoregressive image generation with continuous tokens at scale." arXiv preprint arXiv:2508.10711 (2025).\
[4] Ke, Guolin, and Hui Xue. "Hyperspherical latents improve continuous-token autoregressive generation." arXiv preprint arXiv:2509.24335 (2025).\
[5] Leng, Xingjian, et al. "Repa-e: Unlocking vae for end-to-end tuning of latent diffusion transformers." Proceedings of the IEEE/CVF International Conference on Computer Vision. 2025.\
[6] Yao, Jingfeng, Bin Yang, and Xinggang Wang. "Reconstruction vs. generation: Taming optimization dilemma in latent diffusion models." Proceedings of the Computer Vision and Pattern Recognition Conference. 2025.\
[7] Skorokhodov, Ivan, et al. "Improving the diffusability of autoencoders." arXiv preprint arXiv:2502.14831 (2025).\
[8] Yang, Jiawei, et al. "Latent Denoising Makes Good Tokenizers." The Fourteenth International Conference on Learning Representations.\
[9] Hang, Tiankai, et al. "Fast autoregressive models for continuous latent generation." arXiv preprint arXiv:2504.18391 (2025).\
[10] Fan, Lijie, et al. "Unified autoregressive visual generation and understanding with continuous tokens." arXiv preprint arXiv:2503.13436 (2025).

---

> ### Author Rebuttal · Authors · 2026-03-31
>
> ### Q1: Diagnosing Autoregressive Latent Drift
>
> Direct drift diagnostics would indeed strengthen the mechanistic understanding. Our current evidence characterizes drift indirectly through the relation between tokenizer variance and final FID (Figure 5), where the strong correlation supports the presence of underlying drift. We acknowledge that more direct measurements (e.g., token norm growth and variance evolution over position) would be valuable, and we are working on incorporating such analysis.
>
> We can share one additional observation from our experiments: when using sampled latents as input but predicting the central latent at the output, the model performs better at an early stage of training, but after full convergence at 400 epochs it becomes worse than predicting sampled latents. This suggests that even when the objective appears easier, the train-inference mismatch still accumulates over rollout depth, which is consistent with the drift hypothesis.
>
> ---
>
> ### Q2: Specificity of Gains to $\sigma$-VAE
>
> We have examined several related factors:
>
> 1. Sampled encoder variance mainly addresses the weakness of a fixed-variance design, where reconstruction quality degrades in the low-noise regime because the model overfits to a specific noise level. This is why we use sampled variance.
> 2. Latent noise augmentation mainly improves decoder robustness. As shown in Figure 5, DiT performance remains relatively stable across different variance settings, and the gain is clearly smaller than in the autoregressive case. This supports the view that the variance sensitivity is specific to the autoregressive setting.
> 3. We acknowledge that normalization-based methods and foundation-aligned tokenizer shaping may also address related aspects of the latent space geometry. These techniques are complementary to our approach and may capture overlapping or additional factors beyond variance alone.
>
> Our main contribution is therefore best understood as identifying the broader principle that tokenizer-generator alignment is critical for continuous-token autoregressive generation, with $\sigma$-VAE serving as a simple and effective instantiation. We view $\sigma$-VAE not as the only solution, but as evidence that this alignment problem is real and addressable. We have updated the paper to reflect this framing.
>
> ---
>
> ### Q3: Raster Next-Token Order as Autoregressive Factorization
>
> We agree that autoregressive factorization order is an important design axis. Our focus on raster causal order is motivated by its practical advantages in the multimodal unification setting: it is fully compatible with the KV-cache inference pipeline of standard LLMs, requires no additional order prediction or permutation mechanism, and provides the most natural interface for interleaving discrete text tokens and continuous visual tokens within a single sequence. These properties make raster order the most straightforward choice for integrating continuous generation into existing language model infrastructure.
>
> We note that the tokenizer alignment issue we identify is largely orthogonal to the choice of factorization order — autoregressive drift under recursive rollout would affect any fixed sequential order, not just raster scan. Exploring better factorization strategies (e.g., FAR [9]) is a complementary direction that could be combined with the tokenizer improvements we propose. We have clarified in the revision that our image generation conclusions are established for raster causal AR, and that investigating alternative orders is a promising direction for future work.
>
> ---
>
> ### Q4: Necessity of Diffusion as the Continuous Token Prediction Head
>
> We agree that this is an important question. Our current experiments do not claim that diffusion is the only viable continuous head, and prior work such as GIVT has already explored lighter designs (e.g., Gaussian mixture models). We chose diffusion for several concrete reasons: (1) it can model complex multimodal distributions over continuous latents, which is important when a single token position may correspond to diverse visual content; (2) it naturally accommodates varying levels of latent variance, making it well-suited to work with $\sigma$-VAE where the variance is explicitly controlled; and (3) it is a well-established and reliable predictor for continuous targets, reducing confounding factors in our study of tokenizer alignment.
>
> That said, the core insight of our paper, that tokenizer-generator alignment is critical for continuous-token autoregressive generation, applies regardless of the specific continuous head. A lighter head (e.g., flow matching or a simple Gaussian) would face the same latent drift issue under recursive rollout if the tokenizer is misaligned. We agree that systematically comparing different continuous heads is an interesting direction for future work, and we have added a discussion of this in the revision.

---

> > ### Author Rebuttal · Reviewer_9RyJ · 2026-04-03
> >
> > Thank you for the thoughtful rebuttal. My concerns are partially resolved. The response usefully clarifies the intended contribution: the paper’s main claim is the broader tokenizer–generator alignment principle for continuous-token autoregressive generation, with σ-VAE as a simple and effective instantiation and not necessarily as the uniquely correct solution. This clarification addresses a substantial part of my concern about framing and originality, and I also appreciate the explicit narrowing of scope regarding raster causal AR and the acknowledgment that diffusion is not claimed to be the uniquely necessary continuous head.
> >
> > My main remaining reservation is mechanistic. The rebuttal makes the drift interpretation more plausible, and the additional observation about sampled-latent inputs is helpful, but the evidence is still indirect. The paper continues to support that tokenizer geometry matters for continuous-token AR more strongly than it directly establishes the specific latent-drift mechanism proposed in the paper. In particular, I still think direct diagnostics of rollout-depth drift, token statistics over position, or teacher-forced vs. self-generated prefix divergence would substantially strengthen the central claim.
> >
> > So overall, the rebuttal improves my confidence in the paper’s framing and contribution, but the core mechanistic concern is only partially resolved.

---

> > > ### Author Response · Authors · 2026-04-04
> > >
> > > Thank you very much for the thoughtful follow-up and for the careful reading of our rebuttal.
> > >
> > > We sincerely appreciate your recognition that the paper’s main contribution is better understood as the tokenizer–generator alignment principle for continuous-token autoregressive generation, with σ-VAE as one simple and effective instantiation. We are also grateful that the clarifications on scope and positioning were helpful.
> > >
> > > We also understand your remaining concern on the mechanistic side. We agree this is an important question, although getting fully direct evidence for the mechanism is inherently challenging in this setting. We nonetheless appreciate your suggestions very much, and we believe they point to valuable directions for further strengthening the understanding of this phenomenon.
> > >
> > > Thank you again for the constructive feedback.

---

### Decision · Program_Chairs · 2026-04-30

**Decision:**

Accept (spotlight)

**Comment:**

Final rating: 5: Accept / 4: Weak Accept / 4: Weak Accept / 4: Weak Accept

The paper proposes LatentLM, a unified decoder-only causal Transformer framework that integrates discrete text with continuous modalities like images and speech using next-token diffusion. To stabilize autoregressive generation, the authors introduce $\sigma$-VAE, which utilizes variance sampling to mitigate exposure bias and variance collapse. Reviewers appreciated the clean, unified architecture that maintains compatibility with standard KV-cache infrastructure and avoids the information loss inherent in discrete vector quantization. Initial concerns focused on the mechanistic evidence for $\sigma$-VAE's effectiveness, the limited scope of multimodal understanding evaluations, and the potential inference latency introduced by per-token diffusion steps.

Following the rebuttal, reviewers (9RyJ, iCR5, GFho, 9rCU) noted that the authors successfully addressed several concerns through additional experiments and clarifications. Specifically, the authors provided new text-to-speech ablations validating the importance of tokenizer variance across modalities and included human evaluations showing that their 7B model outperforms strong baselines like Gemini 2.5 Pro Preview TTS. While some reviewers noted that the evidence for the specific "latent-drift" mechanism remains somewhat indirect, they agreed that the "tokenizer-generator alignment" principle is a significant and influential observation for continuous-token autoregression. The authors also committed to toning down broader claims regarding scalability and multi-turn dialogue to better reflect the current experimental scope.

The AC recommends acceptance based on the technical significance of the proposed LatentLM framework and $\sigma$-VAE mechanism, while also taking the constructive author-reviewer discussion into account.